# β-Catenin and FGFR2 regulate postnatal rosette-based adrenocortical morphogenesis

Sining Leng[1,2,10], Emanuele Pignatti [1,3,10], Radhika S. Khetani [4], Manasvi S. Shah[1,3], Simiao Xu[1,3], Ji Miao [1,3], Makoto M. Taketo[5], Felix Beuschlein [6,7], Paula Q. Barrett [8], Diana L. Carlone [1,3,9] & David T. Breault[1,3,9✉]

Rosettes are widely used in epithelial morphogenesis during embryonic development and organogenesis. However, their role in postnatal development and adult tissue maintenance remains largely unknown. Here, we show zona glomerulosa cells in the adult adrenal cortex organize into rosettes through adherens junction-mediated constriction, and that rosette formation underlies the maturation of adrenal glomerular structure postnatally. Using genetic mouse models, we show loss of β-catenin results in disrupted adherens junctions, reduced rosette number, and dysmorphic glomeruli, whereas β-catenin stabilization leads to increased adherens junction abundance, more rosettes, and glomerular expansion. Furthermore, we uncover numerous known regulators of epithelial morphogenesis enriched in β-catenin-stabilized adrenals. Among these genes, we show *Fgfr2* is required for adrenal rosette formation by regulating adherens junction abundance and aggregation. Together, our data provide an example of rosette-mediated postnatal tissue morphogenesis and a framework for studying the role of rosettes in adult zona glomerulosa tissue maintenance and function.

[1] Division of Endocrinology, Boston Children's Hospital, Boston, MA 02115, USA. [2] Division of Medical Sciences, Harvard Medical School, Boston, MA 02115, USA. [3] Department of Pediatrics, Harvard Medical School, Boston, MA 02115, USA. [4] Department of Biostatistics, Harvard T.H. Chan School of Public Health, Boston, MA 02115, USA. [5] Division of Experimental Therapeutics, Graduate School of Medicine, Kyoto University, Yoshida-Konoe-Cho, Sakyo, Kyoto 606-8506, Japan. [6] Department of Endocrinology, Diabetology and Clinical Nutrition, UniversitätsSpital Zürich, Zurich, Switzerland. [7] Medizinische Klinik und Poliklinik IV, Klinikum der Ludwig-Maximilians-Universität München, Munich, Germany. [8] Departments of Pharmacology, University of Virginia, Charlottesville, VA 22947, USA. [9] Harvard Stem Cell Institute, Cambridge, MA 02138, USA. [10]These authors contributed equally: Sining Leng, Emanuele Pignatti. ✉email: david.breault@childrens.harvard.edu

Tissue morphogenesis is critical for embryonic development and organ formation. Conserved morphogenetic processes, such as convergent extension, epithelial folding, branching, and invagination, are widely utilized to facilitate tissue remodeling during development across species[1]. Multicellular rosettes have recently been recognized as a conserved mechanism to efficiently achieve complex cellular reorganization during such processes[2]. Rosettes are defined as five or more cells joined together at a common point of contact[3]. Rosette formation and resolution have been shown to facilitate two-dimensional (2D) tissue elongation[3–5], as well as three-dimensional (3D) changes such as folding and branching during organogenesis[6–9]. Despite the well-established role of rosettes in development, whether they regulate tissue remodeling during postnatal development or in adult tissue homeostasis is largely unknown.

Adherens junctions (AJ) play a crucial role in allowing for coordinated cell movement and shape changes during tissue morphogenesis[10,11]. AJ stability and dynamics can be regulated through cadherin expression and trafficking[12,13], as well as by the mechanical force generated by AJ-associated actin filaments (F-actin) and non-muscle myosin II molecular motors[14,15]. AJ remodeling, mediated by actomyosin contraction, represents a common mechanism underlying rosette formation[2,11], while the mechanism(s) governing rosette resolution are less clear. On the other hand, the extracellular signals governing rosette formation are diverse, and involve the establishment of planar cell polarity and/or apical basal polarity, which in turn leads to a polarized distribution of AJ components, Rho GTPases and their effectors such as the myosin-activating kinase ROCK[2,3,5,16–18]. Whether similar AJ-mediated tissue remodeling mechanisms exist and participate in postnatal and adult tissue regulation remains unknown.

As a constantly self-renewing and highly regenerative tissue, the adrenal cortex is an excellent system to study multicellular dynamics in an adult tissue[19,20]. The adult cortex is organized into morphologically distinct concentric layers called "zones" that start to emerge postnatally as the fetal cortex regresses[21]. The outer most layer, zona glomerulosa (zG), consists of small compact cell clusters that produce aldosterone, a potent mineralocorticoid regulating $Na^+/K^+$ balance and blood pressure[22]. Immediately adjacent to the zG is the zona fasciculata (zF), consisting of larger lipid-rich cells arranged in parallel cords that produce cortisol in humans and corticosterone in mice[23]. A third zone, the zona reticularis (zR), emerges in humans and certain non-human primates before the onset of puberty, which produces androgen precursors. Lineage-tracing studies in mice showed that during adult cortical tissue turnover, zG cells migrate centripetally into the zF, undergoing transdifferentiation and a dramatic morphological remodeling[24]. However, the cellular mechanism(s) that give rise to the unique tissue structures that comprise each zone remain unknown.

The compact cell clusters in the zG have been termed "glomerulus" or "rosette-like" due to their round morphological appearance. However, the exact nature of these structures and their functional significance in the adult adrenal has never been investigated. Given their role in mediating tissue morphogenesis, one appealing hypothesis is that rosette formation and resolution underlie the establishment and remodeling of zone-specific tissue morphology. These questions cannot be addressed without a detailed understanding of the adrenal glomerulus structure and the mechanisms governing its formation.

Here, we present the first detailed analysis of adrenal glomerular morphology. Using confocal imaging and 3D reconstruction, we show that glomeruli in the adult zG are interconnected globular structures enwrapped in a Laminin β1-rich BM. Within each glomerulus, zG cells organize into multicellular rosettes through AJ-mediated membrane constriction. During the first few weeks after birth, rosette formation occurs during a previously unknown process of glomerular morphogenesis. Using genetic loss- and gain-of-function mouse models, we find β-catenin is required for intact rosette structure and that constitutive β-catenin stabilization results in zG morphological expansion, along with increased rosette frequency. Furthermore, using RNA sequencing, we discover that expression of genes associated with epithelial morphogenesis is strongly enriched in β-catenin-stabilized adrenals. Among these genes, we specifically show that *Fgfr2* is essential for rosette formation through modulation of AJ aggregation and abundance. Finally, we find that mice with abnormal rosette morphology exhibit altered zG physiological function, suggesting that appropriate cellular organization within zG rosettes is crucial for normal aldosterone production. Our mechanistic findings suggest that similar principles that govern embryonic development and organogenesis can be utilized to facilitate dynamic tissue remodeling during postnatal development and adult tissue homeostasis.

## Results

**Adult adrenal glomeruli consist of multicellular rosettes.** The term "glomerulus" has been used to describe the compact round cell clusters in the zG. However, what exactly defines these structures has never been formally established. To facilitate the structural definition of the adrenal glomerulus, we performed immunofluorescence staining on 100 μm-thick adrenal slices. Use of thick sections allows for better spatial resolution and preservation of tissue morphology. We co-stained for the Gq alpha subunit (Gαq), a marker of mature zG cells, and classic basement membrane (BM) components such as laminin and type IV collagen. We find laminin subunit β1 (Lamb1) is strongly enriched in the zG (Fig. 1a), consistent with a previous report[25]. Type IV collagen, as marked by Col4a1, is also present in the BM of the zG, as well as the zF (Supplementary Fig. 1a). From a 2D cross-sectional viewpoint, glomeruli are distinct clusters of zG cells surrounded by a Lamb1-rich BM (Fig. 1a). Using 3D reconstructions from confocal z-stack images spanning 50–100 μm depth, we find glomeruli to be globular structures tightly packed into the zG domain that are interconnected through small openings in the BM (Supplementary Movie 1). In addition, we find each glomerulus in 3D contains 15 ± 5 (mean ± standard deviation) zG cells and those at the boundary between the zG and zF appear "half-open" allowing direct contact between zG and zF cells (Supplementary Movie 1). Finally, as in other epithelial tissues, we find that the BM in the adrenal cortex also serves as a barrier between cortical cells and the surrounding vasculature (CD31+) and mesenchyme (Vimentin+) (Supplementary Fig. 1b, c).

Next, we examined the cellular organization within each glomerulus. β-catenin is an integral component of the AJ and the key signal transducer of the canonical Wnt pathway[26]. Consistent with previous reports[27–29], we find that β-catenin is strongly enriched in the zG (Fig. 1b). In addition to a nuclear localization, β-catenin also marks the zG cell membrane, especially along cell-cell contacts (Fig. 1b), indicating the presence of AJs. Notably, we find that within each Lamb1-outlined glomerulus, zG cells are connected at a common membrane center (Fig. 1b, arrowhead). Thus, the structure fulfills the basic geometrical definition of a multicellular rosette, that is five or more cells joined at a common membrane contact point[3]. To further study rosette conformation, we followed the topography of individual cells within a rosette using confocal z-stack images and 3D reconstruction. We find that zG rosettes typically contain 10–15 cells that are connected at

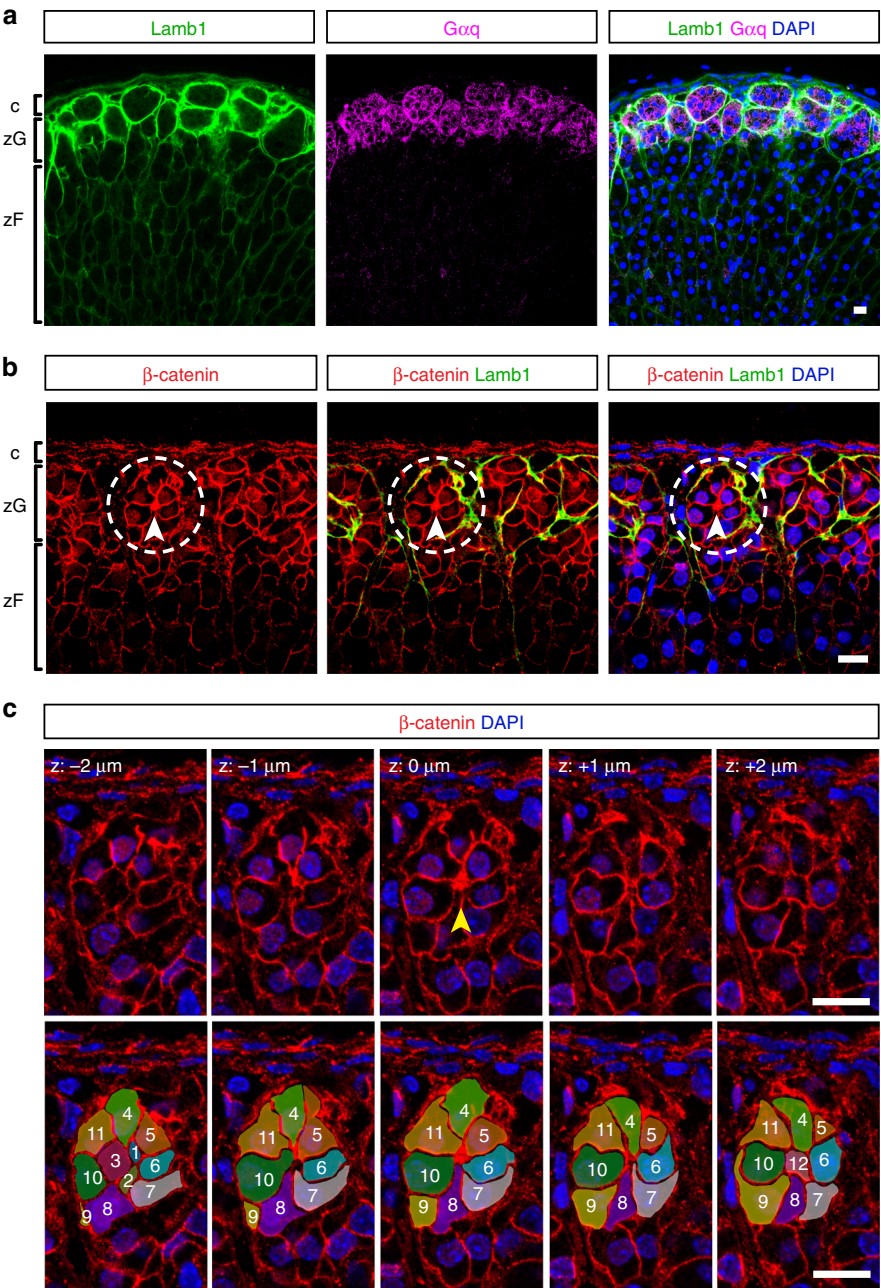

**Fig. 1 Adult adrenal glomeruli consist of multicellular rosettes. a** Laminin β1 (Lamb1, green) marks the basement membrane surrounding distinct clusters of zG cells (Gαq+, magenta), defining the outline of individual glomerulus. **b** Cells within each glomerulus organize into rosettes. Representative image of an adult adrenal slice stained for Laminin β1 (Lamb1, green) and β-catenin (red). Dashed circles highlight a rosette example. Arrowheads point to rosette center. **c** Top, confocal z-stack images of the rosette encircled in (**b**), showing β-catenin (red) and nuclei (DAPI, blue). Z step size is 1 μm. Arrowhead points to the rosette center. Bottom, tracing of cells (pseudo-colored and numbered 1–12) participating in the rosette shown in top panel. c capsule, zG zona glomerulosa, zF zona fasciculata. DAPI (blue) marks nuclei. All bars, 10 μm.

a single vertex located in the center of the structure (Fig. 1c; Supplementary Movie 2). In addition, we observe a roughly one to one relationship between rosettes and glomeruli. However, some larger glomeruli contain more than one rosette while some smaller ones had no rosettes. Surprisingly, zG cells appear to lack a clear apical–basal polarity. While each cell has a basal contact with the BM, no apical domain can be identified either geometrically or with traditional apical markers such as ZO-1, aPKC, and Par3 (data not shown). Together, these data provide a 3D description of the adrenal glomerulus structure, in which non-polarized cells form rosettes that are present in the adult zG.

The mouse zG is a heterogeneous layer comprised of differentiated Cyp11b2+ (aldosterone-producing) cells and less mature Cyp11b2− progenitor cells[20,30]. To examine whether cellular heterogeneity exists within the zG glomerulus, we stained wild-type adult adrenal sections for Cyp11b2 and β-catenin, a nondiscriminatory marker of all zG cells[19]. We find that a typical glomerulus contains both Cyp11b2+ and Cyp11b2− cells (Supplementary Fig. 2a). Quantitative analysis reveals a wide distribution in the percentage of cells expressing Cyp11b2 per glomerulus, with the majority ranging from 20 to 70% (Supplementary Fig. 2b). These data indicate that the adrenal

glomerulus largely represents a morphological structure where both functional zG cells and their precursors reside.

**Rosette centers have condensed AJ**. Rosettes commonly form through AJ-mediated membrane constriction[2]. To investigate whether zG rosettes may form by a similar mechanism, we examined the localization of AJ components in zG rosettes, including cadherins and filamentous actin (F-actin). Several type I and II classical cadherins are expressed in the adult adrenal gland (ENCODE, GEO accession number: GSM900188). Using available antibodies, we find that both N-cadherin (Cdh2) and K-cadherin (Cdh6) are enriched on the cell membrane of zG cells and show relatively weaker expression on zF cells (Supplementary Fig. 3a, b). In contrast, E-cadherin (Cdh1) is not found in the adrenal cortex (Supplementary Fig. 3c). F-actin, together with non-muscle myosin II, form the actomyosin network essential for mechano-transduction and AJ remodeling[11]. Phalloidin staining reveals F-actin has a punctate appearance on zG cell membranes and these punctae form larger aggregates at rosette centers (Fig. 2a–c, Supplementary Fig. 3a, b). Furthermore, we find that β-catenin, N-cadherin, and K-cadherin all colocalize with F-actin aggregates at rosette centers and at small dispersed punctae (Fig. 2a–c, Supplementary Fig. 3a, b), suggesting that these punctae represent AJs and associated actomyosin structures. In comparison, F-actin and AJ components are weakly present on zF cell membranes and do not form punctae (Supplementary Fig. 3a, b), suggesting that AJs and their associated actomyosin may play a less prominent role in zF cell–cell adhesion and tissue morphology.

To further confirm the identity of these junctions and to understand the ultrastructural details of rosette centers, we performed transmission electron microscopy. As previously reported[31], adult zG cells can be distinguished by numerous mitochondria and few lipid droplets, in contrast to the lipid-rich zF cells (Fig. 2d, left). We find that zG cells are in contact with the rosette center via narrowing membrane protrusions (Fig. 2d, middle). Strikingly, stereotypical AJ structures with electron-dense actin filaments appear as linear aggregates at zG cell–cell contacts (Fig. 2d, right). This type of AJ arrangement is in sharp contrast with a "tripartite" apical junction complex of an epithelial cell with apical–basal polarity, where tight junction, adherens junction, and desmosome appear in a sequence[32]. Together, these data confirm the presence of AJs in the zG, and show that AJs condense at rosette centers, suggesting a potential role for AJ-mediated membrane constriction in the formation of rosettes in the zG.

To determine whether zG rosettes are evolutionarily conserved, we performed immunostaining for AJ components on thick paraffin sections of adrenals from human and a Peruvian mouse strain, previously considered not to have a morphological zG[33]. Interestingly, we find that zG cells in both human and Peruvian mouse adrenals are organized in rosettes, highlighting a potentially conserved function for these structures (Supplementary Fig. 3d, e). However, due to the unavailability of non-paraffin-embedded samples (required for Phalloidin staining), we could not confirm whether these adrenals have similar F-actin-rich AJ aggregates at rosette centers.

**Rosette formation underlies postnatal glomeruli morphogenesis**. Since rosettes are known to mediate morphogenesis during development, we next investigated what role rosettes may play in establishing glomerular morphology. To understand when and how glomeruli form in the zG, we examined their morphology using the BM marker Lamb1 at postnatal days 0, 12, and 3 weeks (P0, P12, and 3 wk) and compared them to the zG of 6-week-old adult mice (6 wk). Interestingly, we find that mature glomeruli arise through a gradual morphogenetic process (Fig. 3a, b). In neonatal adrenals (P0), the zG consists largely of flat sheet-like domains, which over time invaginate, fold and segment, leading to the formation of mature glomeruli with well-defined boundaries (Fig. 3a, b). To quantify this process, we manually traced Lamb1 staining to identify individual developing glomeruli and conducted morphometric analyses (Supplementary Fig. 4a). Glomerular roundness, a shape descriptor indicating closeness to a perfect circle, significantly increases from P0 to adulthood (Fig. 3d). By contrast, glomerular 2D cross-sectional area measurement reveals no significant difference among all stages (Fig. 3e), suggesting increase in cell mass is not a main contributor to glomerular morphogenesis.

To understand whether rosette formation plays a role in glomerular morphogenesis, we examined F-actin distribution during the same time course (Fig. 3a–c). At P0, F-actin appears as small dispersed punctae. At P12, F-actin becomes enriched along zG cell-cell contacts, forming lines. At 3 week, F-actin begins to aggregate, and by 6 weeks of age forms distinct clusters that mark mature rosette centers (Fig. 3b, c). To confirm these findings, we measured the size and length of F-actin punctae at each stage using 3D reconstruction of confocal z-stack images (Supplementary Fig. 4b). We find that F-actin punctae peak in length at P12 and become shorter in the adult, without any significant change in their size as calculated by 3D volume (Fig. 3f, g). Together, our data show that adult glomerular morphology matures postnatally, concurrent with rosette formation, suggesting rosette formation may provide the morphogenetic force underlying zG maturation.

**Effects of dietary salt on rosette and glomerular morphology**. A major function of the zG is to promote sodium reabsorption[23]. In turn, dietary sodium restriction increases zG cell proliferation through the action of angiotensin II, whereas sodium loading inhibits zG proliferation[34,35]. To determine if varying dietary sodium levels impacts zG glomerular morphology, we treated adult mice for 1 week with high salt (HS), low salt (LS), or normal salt (NS) diet. Using Lamb1 staining to identify glomerular boundaries, we then examined the zG morphology of these mice (Supplementary Fig. 5a). Compared to NS diet, the glomeruli of mice in the HS group appear flatter and smaller, as confirmed by both reduced glomerular roundness and 2D cross-sectional area (Supplementary Fig. 5b, c). On the other hand, mice in the LS group appear to have more elongated and larger glomeruli, also as reflected by reduced glomerular roundness and an increase in 2D cross-sectional area (Supplementary Fig. 5b, c). Quantification of rosette numbers reveals no significant change, although a moderate trend towards an increase in rosette frequency was observed between the HS and LS groups (Supplementary Fig. 5d). Together, these data indicate that varying dietary sodium levels can have a pronounced impact on zG glomerular morphology.

**β-Catenin regulates rosette and glomerular morphology**. β-catenin is an integral AJ component, linking cadherin molecules to the actin cytoskeleton through binding α-catenin[36,37]. In addition, β-catenin is the key signal transducer of the canonical Wnt pathway, which has previously been linked to zG differentiation[38]. To begin to understand the role of β-catenin and AJs in zG rosette formation, we first tested whether loss of β-catenin results in altered rosette structures and glomerular morphology. For these studies, we generated mice with β-catenin conditional knockout in zG cells ($AS^{Cre/+} :: Ctnnb1^{fl/fl}$, referred to as βCat-LOF). Since AS-Cre expression begins at birth and is expressed in aldosterone-producing zG cells throughout adult life[24], we analyzed adrenals at 10–13 weeks of age to evaluate the cumulative effect of β-catenin deletion. In both female and male βCat-LOF

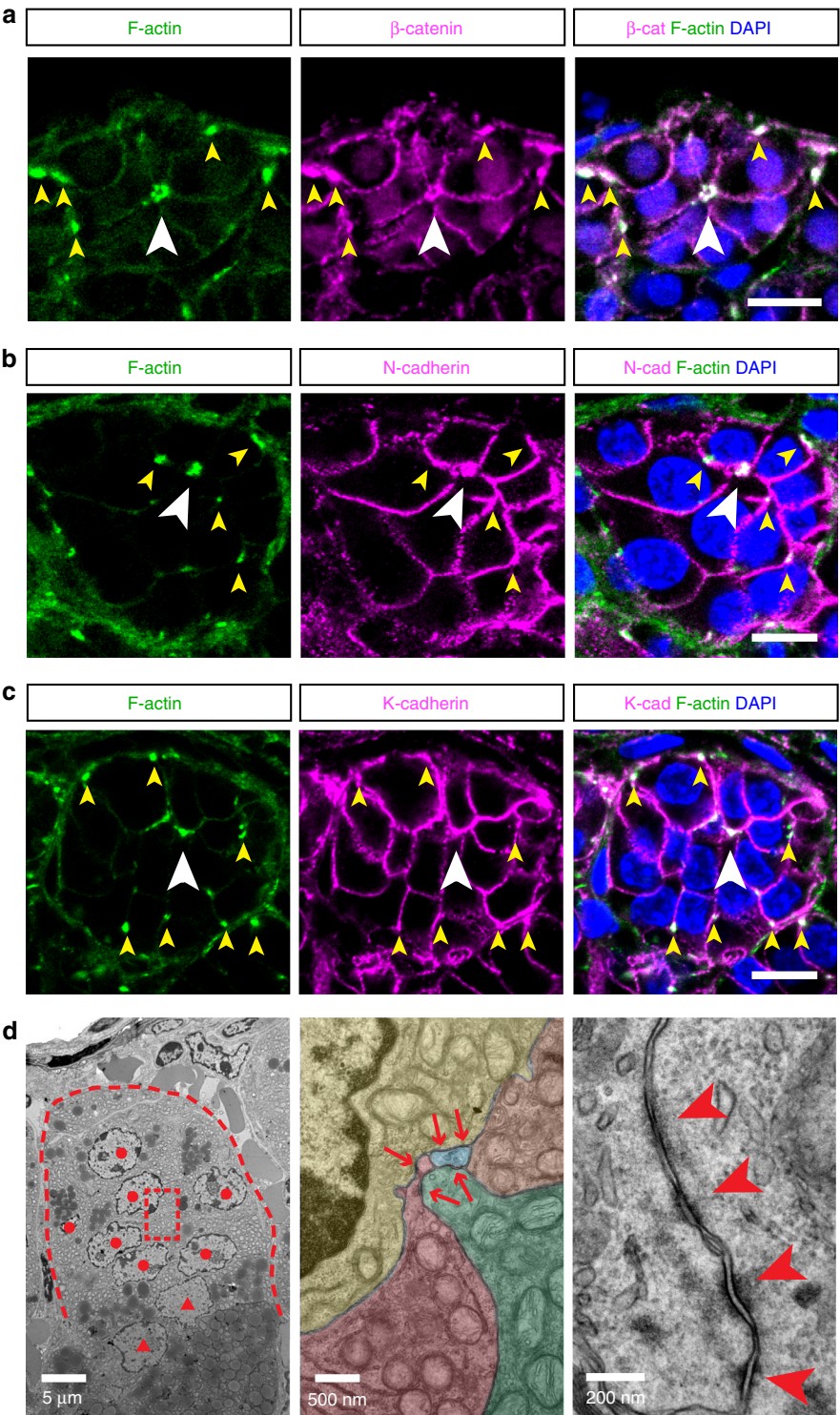

**Fig. 2 Adherens junction components are enriched at rosette centers. a–c** Colocalization of F-actin (green) and β-catenin (β-cat, magenta), N-cadherin (N-cad, magenta), and K-cadherin (K-cad, magenta) at rosette centers (white arrowheads) and smaller F-actin punctae (yellow arrowheads). DAPI (blue) marks nuclei. All bars, 10 μm. **d** Transmission electron micrographs showing adherens junctions in the zG. Left, red dashed line marks the boundary of a glomerulus. Red dots denote nuclei of zG cells. Red triangles denote nuclei of zF cells. Boxed area is shown at a higher magnification in the middle panel where individual cells are pseudo-colored. Red arrows point to adherens junctions. Right, an example of aggregating adherens junctions (red arrowheads). Bar sizes are indicated in each image.

adrenals, we find that glomeruli are significantly less round, have smaller cross-section area and contain less cells (Fig. 4a–d). Overall, there is a trend toward a decrease in zG-layer thickness (Fig. 4f). In addition, F-actin punctae are largely absent and the overall rosette frequency is reduced in βCat-LOF adrenals (Fig. 4e–g). Furthermore, we find that K-cadherin is also absent from β-catenin-deleted zG cell membranes (Supplementary Fig. 6), consistent with the role of β-catenin in maintaining AJ

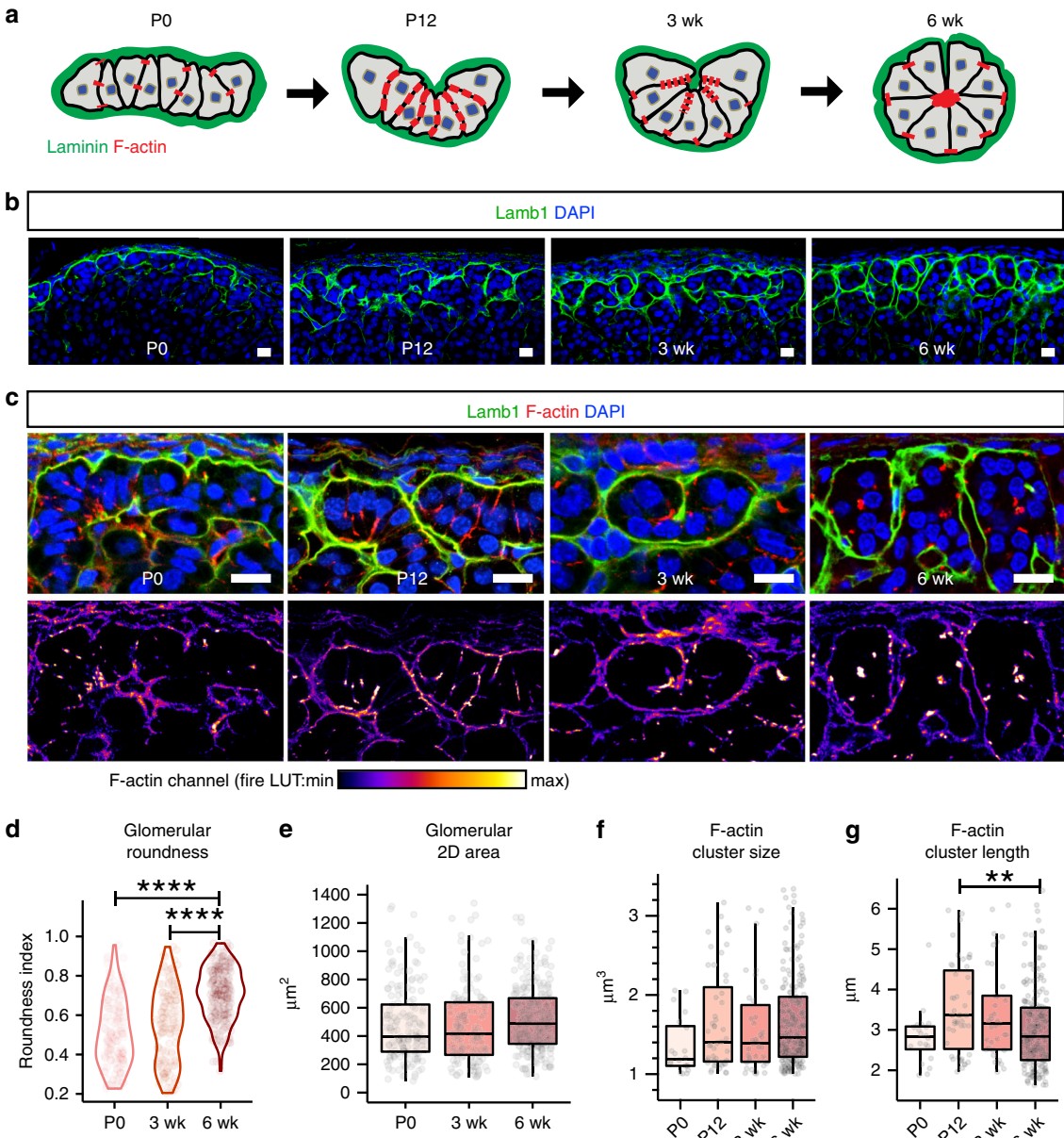

**Fig. 3 Rosette formation underlies postnatal glomerular morphogenesis. a** Schematic of glomerular morphogenesis. **b, c** Time course of postnatal glomeruli morphogenesis. **b** Lamb1 (green) marks the basement membrane surrounding the developing glomeruli. **c** F-actin (red) distribution within the developing glomeruli (outlined by Lamb1 in green) at indicated postnatal stages. DAPI (blue) marks nuclei. Bars, 10 μm. Lower panels show F-actin channel alone in fire LUT. **d** Measurement of glomerular roundness at indicated stages. Kruskal–Wallis test, $P < 0.0001$; P0 versus 6 week, Dunn's test, ****$P < 0.0001$; 3 week versus 6 week, Dunn's test, ****$P < 0.0001$. **e** Measurement of glomerular cross-sectional area at indicated stages. For **d** and **e**, at least three animals from each time point and at least fifty glomeruli per animal were examined. **f** Measurement of F-actin cluster size at indicated stages. **g** Measurement of F-actin cluster length at indicated stages. Kruskal–Wallis test, $P < 0.01$; P12 versus 6 week, Dunn's test, **$P < 0.01$. For **f** and **g**, at least two animals from each time point and at least ten F-actin punctae per animal were examined. For all measurements, first a Kruskal–Wallis rank sum test was performed, and if significant ($P < 0.05$), followed by Dunn's multiple comparison test for each pair. In all boxplots, box boundaries represent the 25th and 75th percentiles, whiskers represent the 5th and 95th percentiles, center lines represent median. Source data are provided as a Source Data file.

stability. Together, these data show β-catenin is required for intact rosettes and normal glomerular morphology.

The cytosolic pool of β-catenin is modulated by a destruction complex and the proteasome under the control of canonical Wnt signaling[39]. Accumulation of cytosolic β-catenin can affect cell–cell adhesion by saturating cadherin binding or by transcriptional control of genes regulating cell adhesion[40–42]. To test whether β-catenin stabilization via exon 3 deletion alters rosettes and glomerular morphology, we generated zG-specific β-catenin gain-of-function mice ($AS^{Cre/+}$, $Ctnnb1^{fl(ex3)/+}$; referred

to as βCat-GOF). We found that by 10 weeks of age, female βCat-GOF adrenals contain a markedly expanded zG characterized by a Lamb1-rich BM (Fig. 5a, Supplementary Fig. 7b). Compared to female controls, βCat-GOF glomeruli that are more elongated, bigger, and contain more cells (Fig. 5b–d). In the expanded zG of βCat-GOF adrenals, phalloidin staining (F-actin) shows the presence of rosettes (Fig. 5e), and a significant increase in rosette frequency (Fig. 5f). Adrenals from male βCat-GOF mice also exhibit zG expansion, albeit to a milder extent than females (Supplementary Fig. 7a, b). However, we find that the shape and

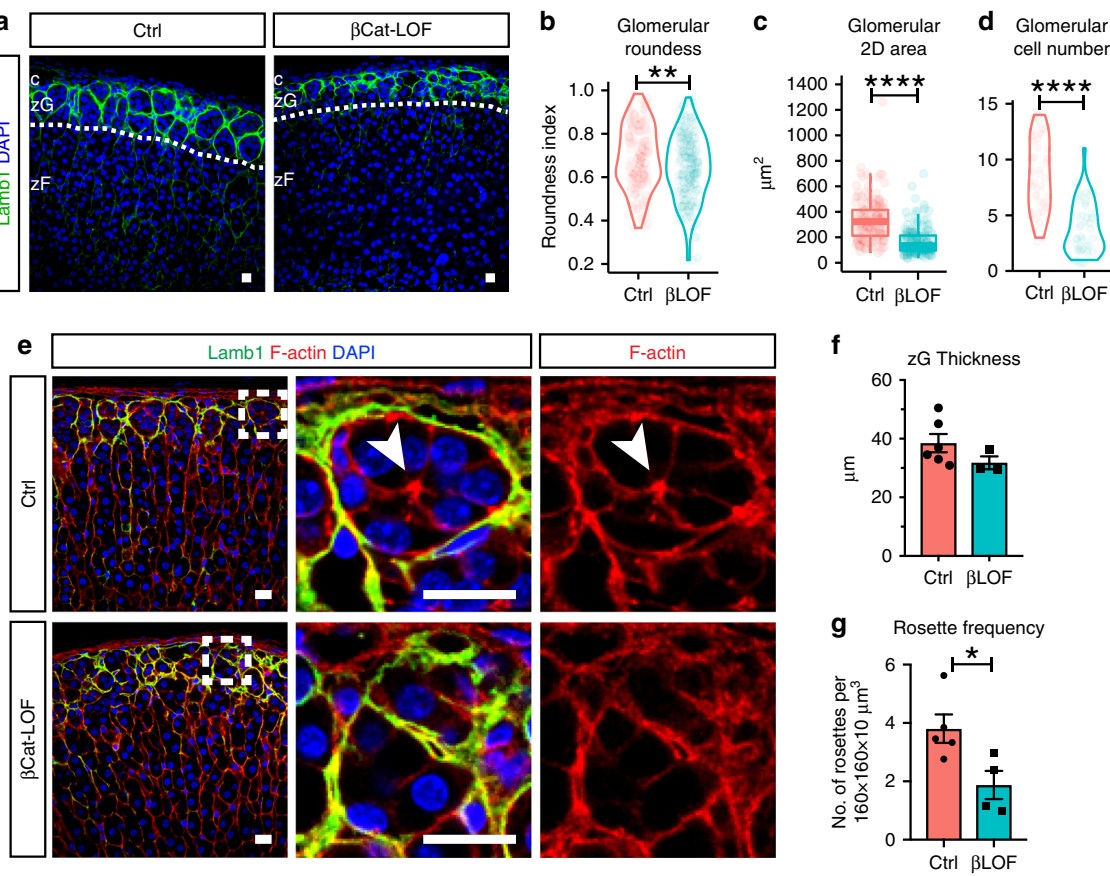

**Fig. 4 β-catenin deletion leads to disrupted rosette and glomerular morphology. a** Lamb1 staining (green) showing glomerular morphology in β-catenin loss-of-function (βLOF) adrenals compared to controls (Ctrl). DAPI (blue) marks nuclei. Dotted line demarcates the zG/zF boundary. c capsule, zG zona glomerulosa, zF zona fasciculata. **b** Measurement of glomerular roundness. Mann–Whitney's nonparametric test, **$P < 0.01$. **c** Measurement of glomerular cross-sectional area. Mann–Whitney's nonparametric test, ****$P < 0.0001$. For **b** and **c**, at least four animals (2 females and 2 males) and 30 glomeruli per animal were examined. **d** Measurement of glomerular cell number. Mann–Whitney's nonparametric test, ****$P < 0.0001$. Four mice per group and at least ten glomeruli per mouse were examined. **e** F-actin staining (red) reveals loss of rosette structures in βCat-LOF adrenals. Glomerular boundary is outlined by Lamb1 (green) staining. DAPI (blue) marks nuclei. Boxed areas in left panels are shown in higher magnification on the right. Arrowhead points to rosette center. **f** Measurement of zG thickness. **g** Rosette frequency calculated as rosette number per $160 \times 160 \times 10\ \mu m^3$ cortical area. Student's $t$ test, *$P < 0.05$. $N = 5, 4$ mice. For each animal, data from three different cortical areas were averaged. All bars, 10 μm. In all bar plots, error bars represent SEM. In all box plots, box boundaries represent the 25th and 75th percentiles, whiskers represent the 5th and 95th percentiles, center lines represent median. Source data are provided as a Source Data file.

size of glomeruli in male βCat-GOF adrenals are similar to controls (Supplementary Fig. 7c, d). In addition, a more modest increase in rosette frequency is detected in male βCat-GOF adrenals compared to females (Supplementary Fig. 7e). Further, we find an enrichment in AJ components including N-cadherin and K-cadherin in the expanded zG in both female and male βCat-GOF adrenals (Supplementary Fig. 7f, g), consistent with their potential role in rosette formation. We think the observed sexual-dimorphic phenotype is likely due to the intrinsic difference in the turnover rate of adrenocortical tissue between male and female mice[43,44]. However, the basic features of glomerular morphology are similarly present in the expanded zG in both sexes. Finally, to assess the impact of zG expansion on physiological function, we measured aldosterone output from control and βCat-GOF mice, which reveals significantly higher plasma aldosterone levels in βCat-GOF mice (Fig. 5g). Together, these data show that β-catenin stabilization drives zG morphological expansion, which is accompanied by an increase in rosette frequency and an increase in aldosterone production.

**Transcripts enriched in βCat-GOF regulate morphogenesis.** To investigate what mechanism(s) may mediate the effects of β-

catenin on zG morphology, we compared transcriptome changes between wild type and βCat-GOF adrenals by RNA sequencing. Differential expression analysis reveals 306 upregulated and 219 downregulated genes in βCat-GOF adrenals compared to controls (Fig. 6a). To understand the functional significance of these changes, we performed enrichment analysis of Gene Ontology Biological Process terms using DAVID (NIH, v6.7). We find top-enriched terms include morphogenesis of an epithelium, Wnt signaling pathway, regulation of cell adhesion, and adherens junction assembly (Fig. 6b), suggesting extensive transcriptional changes underlying zG morphological expansion. On the other hand, GO terms represented by downregulated genes are predominantly involved in innate immune regulation, the significance of which in the adrenal is currently unknown. Among the upregulated Wnt signaling pathway genes are well-established β-catenin targets, such as *Axin2*, *Tcf7*, *Lef1*, *Notum*, *Nkd1*, *Apcdd1*, confirming the expansion of the β-catenin transcription activity domain (Supplementary Fig. 8a–c). Interestingly, several non-canonical Wnt pathway members are also upregulated, including *Nfatc4*, *Daam2*, *Ror2*, and *Dact1*, suggesting their potential role in the regulation of rosette and glomerular morphology (Supplementary Fig. 8a–c).

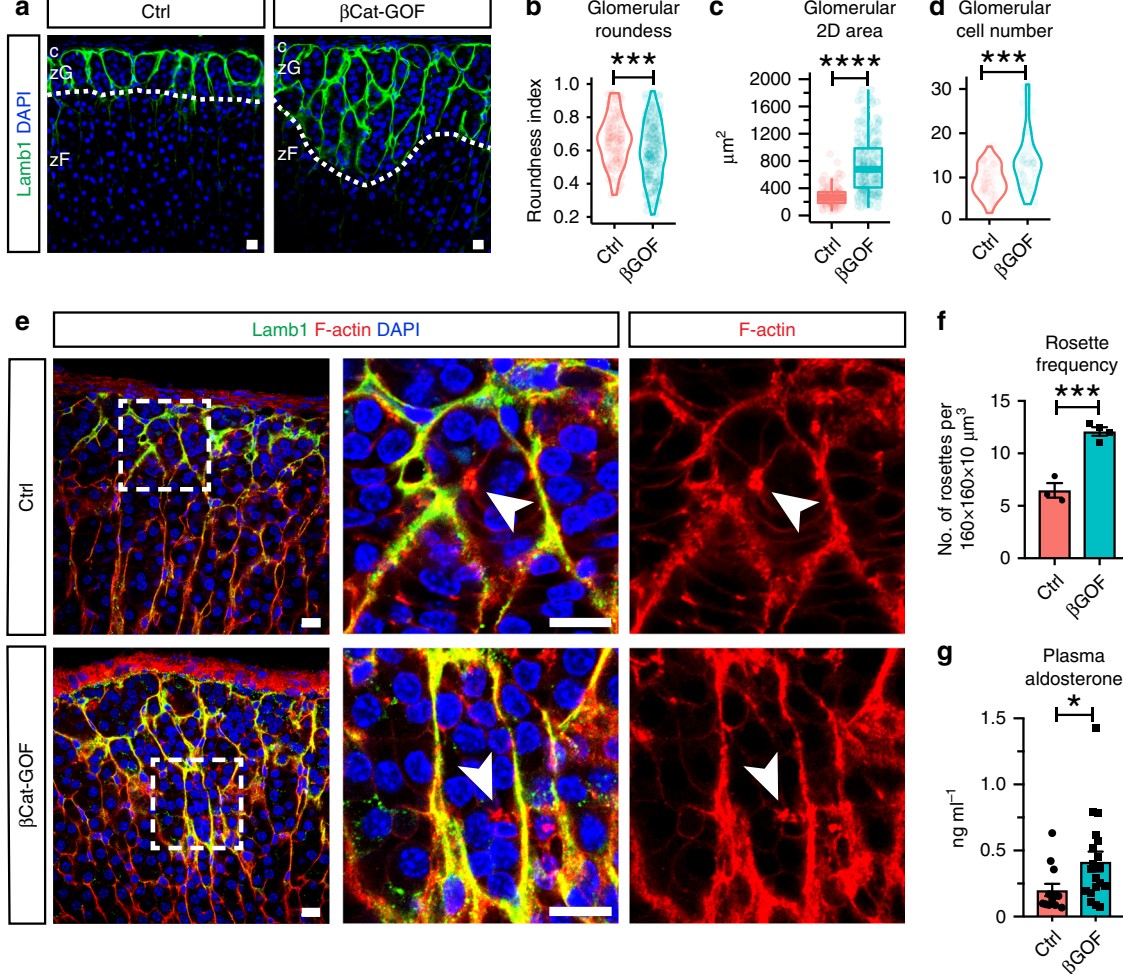

**Fig. 5 β-catenin stabilization results in zG expansion and increased rosette frequency. a** Lamb1 staining (green) in β-catenin gain-of-function (βGOF) adrenals of female mice and controls (Ctrl). DAPI (blue) marks nuclei. Dotted line demarcates the zG/zF boundary. c capsule, zG zona glomerulosa, zF zona fasciculata. **b, c** Measurement of glomerular roundness and cross-sectional area. Mann–Whitney's non-parametric test, ***$P < 0.001$; ****$P < 0.0001$. Three animals from each group and at least 30 glomeruli per animal were examined. **d** Measurement of glomerular cell number. Mann–Whitney's nonparametric test, ***$P < 0.001$. Three mice per group and at least ten glomeruli per mouse were counted. **e** F-actin staining (red) shows rosettes in expanded zG of βCat-GOF adrenals in females. Lamb1 (green) outlines glomerular boundary. DAPI (blue) marks nuclei. Boxed areas are shown in higher magnification on the right. Arrowheads point to rosette centers. **f** Rosette frequency calculated as rosette number per $160 \times 160 \times 10$ μm³ z-stack cortical area. Student's t test, ***$P < 0.001$, $N = 3$, 4 mice. For each animal, data from three different cortical areas were averaged. **g** Plasma aldosterone levels are elevated in female βCat-GOF mice compared to controls. Mann–Whitney's nonparametric test, *$P < 0.05$, $N = 13$, 19 mice. All bars, 10 μm. In all bar plots, error bars represent SEM. In all box plots, box boundaries represent the 25th and 75th percentiles, whiskers represent the 5th and 95th percentiles, center lines represent median. Source data are provided as a Source Data file.

Among the genes representing the GO term morphogenesis of an epithelium, we find potential regulators of rosette formation, such as *Fgfr2* and *Shroom3* (Supplementary Fig. 8b). In the developing lateral line primordium of the zebrafish *Danio rerio*, FGF signaling plays an important role in rosette formation by inducing *Shroom3* expression[16]. Shroom3 is an actin-binding protein that recruits ROCK to cell junctions where ROCK triggers actomyosin contraction[7,45]. We find increased FGFR2 protein in total adrenal extracts from βCat-GOF mice (Fig. 6c, d). Since the whole adrenal was used in the RNAseq experiment, we examined the localization of *Fgfr2* and *Shroom3* transcripts in adrenal cortex using single-molecule in situ hybridization (RNAscope). We find that both transcripts are expressed by cortical cells (Fig. 6e). *Fgfr2* appears to be slightly more abundant in the zG than in the zF, whereas *Shroom3* expression is highly specific to the zG (Fig. 6e, left). Furthermore, we find that both genes are expressed in the expanded zG region in βCat-GOF adrenals (Fig. 6e, right).

Together, our data reveal numerous transcriptional changes that correlate with a morphologically expanded zG driven by β-catenin stabilization. Many of these genes have well-established roles in regulating epithelial morphogenesis, cell adhesion, and junctional dynamics, suggesting they may regulate zG rosette and glomerular morphology.

**FGFR2 mediates rosette formation via AJ regulation.** Because FGFR2 and Shroom3 are important regulators of rosette formation during development in the zebrafish and the expression levels of each are increased in βCat-GOF adrenals, we investigated whether FGFR2 has a role in zG rosette formation. To this end, we generated zG-specific *Fgfr2* loss-of-function mice (AS^{Cre/+}, *Fgfr2*^{fl/fl}; referred to as Fgfr2-LOF) and observed in both male and female adult mice a severely disrupted zG morphology with a trend toward a decrease in zG thickness (Fig. 7a, b). Morphometric analyses show that glomeruli in Fgfr2-LOF adrenals have

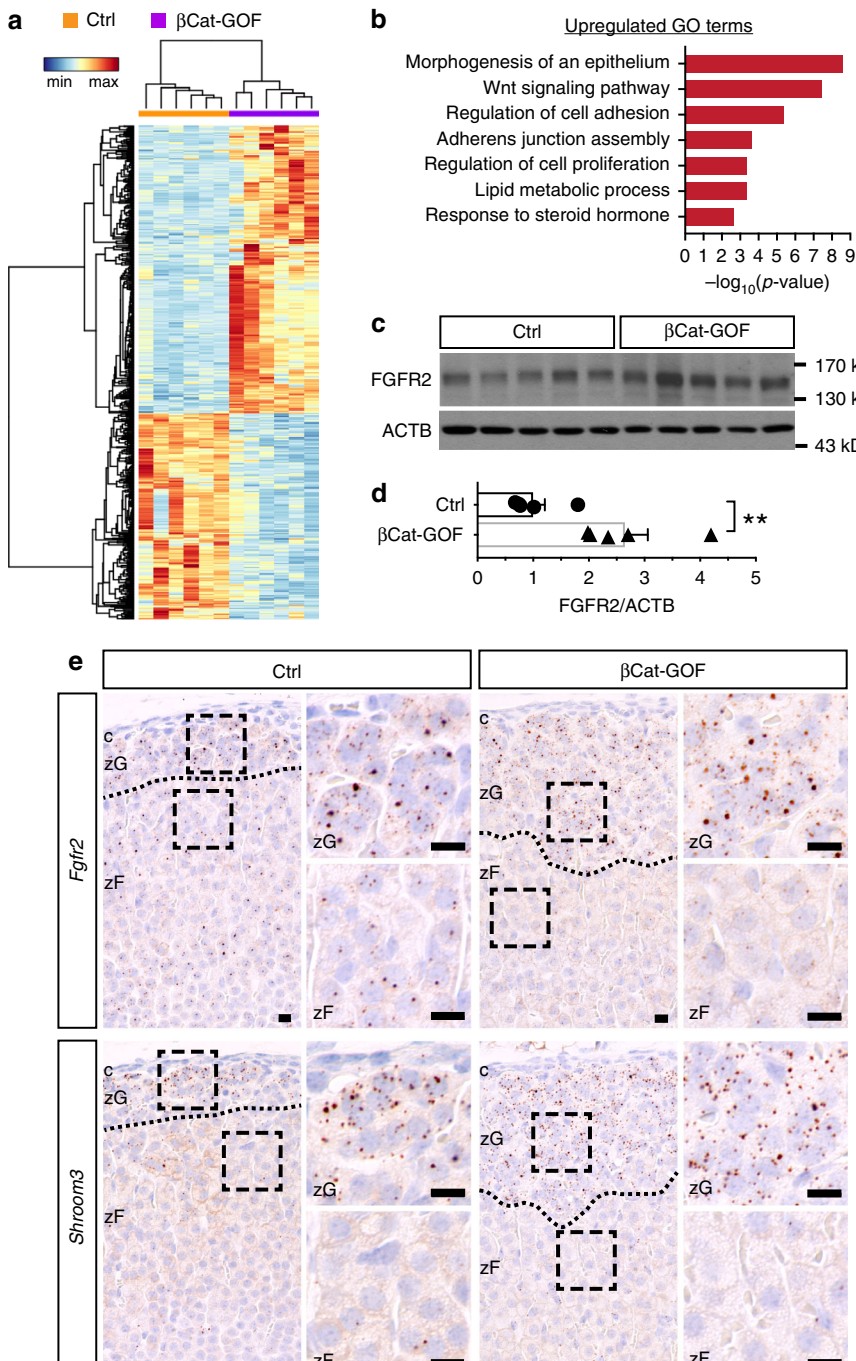

**Fig. 6 Transcripts enriched in βCat-GOF regulate morphogenesis. a** Heatmap of differentially expressed genes between control (Ctrl) and βCat-GOF adrenals based on normalized counts. Fold change > 1.4, adjusted *P* value < 0.05, base mean expression > 100 were used as cut-off criteria. Dendrograms represent hierarchical clustering of samples (top, Spearman correlation) and genes (left, Pearson correlation) using the average linkage method. *N* = 6, 6 female mice. **b** Gene Ontology terms (Biological Processes) enriched in βCat-GOF adrenals. **c** Western blots showing increased FGFR2 in βCat-GOF adrenals compared to Ctrl. **d** Normalized luminescence intensity of (**c**). Student's *t* test, **P* < 0.01, *N* = 5, 5 female mice. **e** Single-molecule in situ hybridization (RNAscope) of *Fgfr2* and *Shroom3* in Ctrl and βCat-GOF female adrenals. Blue, hematoxylin counterstain. Dotted line demarcates zG/zF boundary; boxed areas are shown at higher magnification on the right. c capsule, zG zona glomerulosa, zF zona fasciculata. Bars, 10 μm. In all bar plots, error bars represent SEM. Source data are provided as a Source Data file.

no significant change in roundness (Fig. 7c), but smaller cross-sectional area (Fig. 7d). Notably, in Fgfr2-LOF adrenals, F-actin punctae are present but fail to form aggregates at rosette centers; instead, they are smaller and dispersed (Fig. 7e). Accordingly, rosette frequency is markedly decreased in Fgfr2-LOF adrenals (Fig. 7f). Finally, *Shroom3* expression is reduced in Fgfr2-LOF adrenals (Supplementary Fig. 9a), consistent with its role as a

mediator of FGF signaling. These data show that FGFR2 is required for zG rosette formation and proper glomerular morphology, and that a similar Shroom3-dependent mechanism may be required for AJ aggregation and rosette formation in the adrenal zG.

Since FGF signaling is known to promote cell adhesion by regulating cadherin expression[46–48], we next examined the

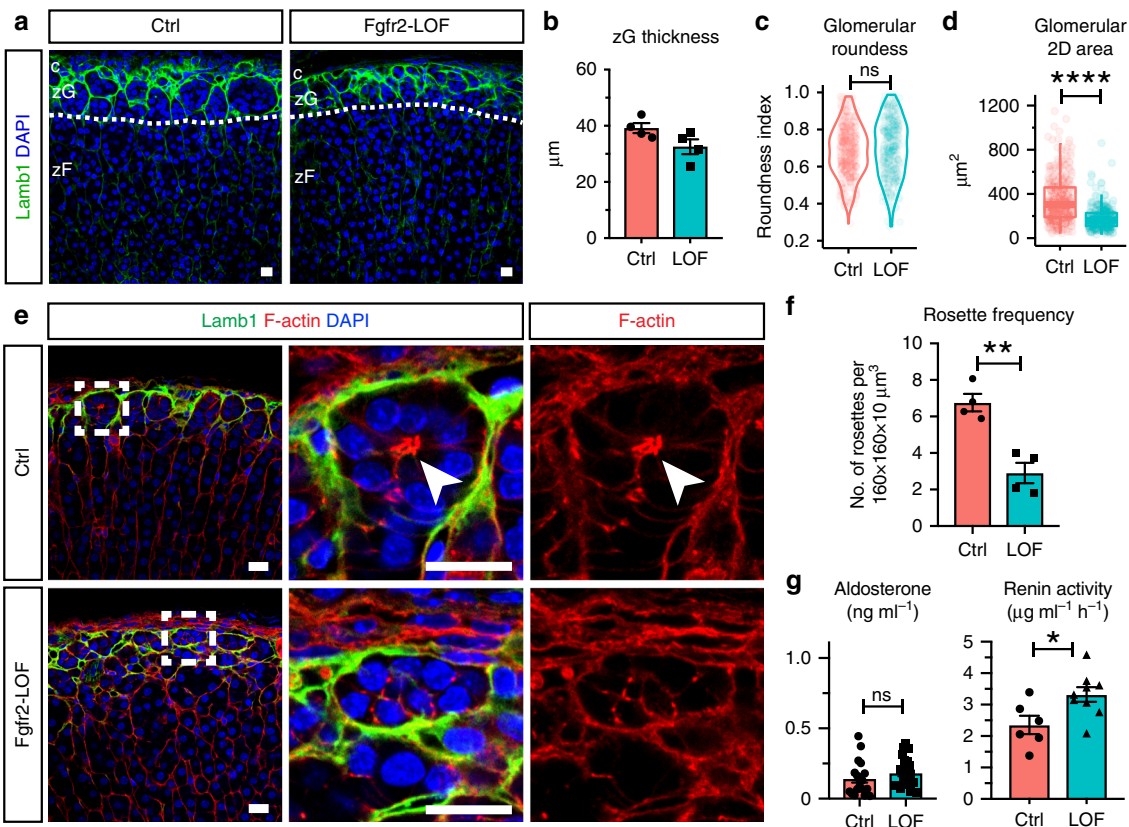

**Fig. 7 Fgfr2 deletion disrupts rosette morphology and zG physiological function. a** Lamb1 staining (green) shows disrupted glomerular morphology in Fgfr2 loss-of-function (LOF) adrenals compared to controls (Ctrl). DAPI marks nuclei (blue). Dotted line demarcates the zG/zF boundary. c, capsule; zG, zona glomerulosa; zF, zona fasciculata. **b** Measurement of zG thickness. $N = 4$, 4 mice. **c, d** Measurement of glomerular roundness and cross-sectional area. Mann–Whitney's nonparametric test, ns not significant, ****$P < 0.0001$. For **c** and **d**, four animals from each group (2 males and 2 females) and at least 30 glomeruli per animal were examined. **e** F-actin staining (red) reveals loss of rosette structures in the glomeruli of Fgfr2-LOF adrenals. Glomerular boundary is outlined by Lamb1 (green). Boxed areas are shown in higher magnification on the right. Arrowheads point to rosette center. DAPI marks nuclei (blue). **f** Rosette frequency calculated as rosette number per $160 \times 160 \times 10$ μm³ z-stack cortical area. Student's t test, **$P < 0.01$, $N = 4$, 4 mice. For each animal, data from three different cortical areas were averaged. **g** Plasma aldosterone level ($N = 18$, 21 mice) and plasma renin activity ($N = 6$, 9 mice) in Ctrl and LOF mice. Mann–Whitney's nonparametric test, ns not significant; *$P < 0.05$. All bars, 10 μm. In all bar plots, error bars represent SEM. In all box plots, box boundaries represent the 25th and 75th percentiles, whiskers represent the 5th and 95th percentiles, center lines represent median. Source data are provided as a Source Data file.

expression levels of N- and K-cadherin. Interestingly, both cadherin transcript levels are significantly reduced in Fgfr2-LOF adrenals (Supplementary Fig. 9a), suggesting FGFR2 regulates zG cell–cell adhesion also by promoting cadherin expression. Consistent with reduced AJ abundance, we notice that β-catenin is lost from the membrane of zG cells in Fgfr2-LOF adrenals (Supplementary Fig. 9b). By contrast canonical Wnt signaling activity remains unaffected in Fgfr2-LOF adrenals. No change in *Axin2* is detected between control and Fgfr2-LOF adrenals (Supplementary Fig. 9c). Similarly, immunostaining of Lef1, an established β-catenin transcriptional target, reveals no difference between control and Fgfr2-LOF adrenals, each exhibiting a zG-specific pattern (Supplementary Fig. 9d). Together, our data support a role for FGFR2 in promoting zG cell–cell adhesion, without directly affecting canonical Wnt signaling activity.

Finally, to assess the impact of Fgfr2-LOF on zG physiological function, we measured aldosterone output from control and Fgfr2-LOF mice. While no significant change in plasma aldosterone levels is observed between Fgfr2-LOF and control mice (Fig. 7g, left), Fgfr2-LOF mice demonstrate a significant increase in plasma renin activity (Fig. 7g, right). Together, these data suggest that activation of the renin–angiotensin

II system in Fgfr2-LOF results in a state of compensated hypoaldosteronism.

## Discussion

Multicellular rosettes represent a widely utilized developmental intermediate employed during tissue morphogenesis. Here, we describe a clear example of bona fide rosette structures present in an adult epithelium outside of the neural system[2]. We find that in mature adrenal glomeruli, 10–15 cells share a common membrane center where their AJs form dense aggregates. The geometry of these rosettes does not fully resemble existing examples in other contexts. Rosettes deriving from planar-cell-polarity (PCP) signals, such as in the Drosophila germband and the vertebrate kidney tubule[3,5,49], resemble a "pinwheel" structure where cells share a common contact point along the entire lateral side. On the other hand, rosettes formed by apical constriction, such as in the zebrafish lateral line primordium and the pancreatic exocrine duct[6,9], resemble a "garlic bulb" shape where cells share a single contact point at the constricted apical surface. Here, our 3D reconstructions clearly demonstrate that zG cells form a single vertex at the center of the rosette. Enwrapped in a BM, zG cells have a clear basal domain where AJ components are absent. However, zG cells have highly irregular shapes and no

clear apical domain can be defined geometrically or molecularly. Hence, we propose that zG rosettes represent a variation on apical constriction, where instead, AJ-mediated membrane constriction seems to be the underlying mechanism.

We find that during postnatal glomerular morphogenesis, rosettes form through dynamic AJ remodeling. Through this process, nascent zG cells are brought together from a flat epithelial sheet into a compact, interconnected network of globular structures. In the mature glomeruli, AJs form dense aggregates at rosette centers. Because of the overlapping timeline of rosette formation and glomerular morphogenesis, we propose a model where rosette formation provides the underlying force driving glomerular morphogenesis. However, we cannot exclude other potential mechanisms contributing to glomerular morphogenesis, such as basal surface remodeling or cell shape changes driven by cell-intrinsic factors[50,51]. Additional unanswered questions pertaining to rosette formation and glomerular morphogenesis, include (1) what mechanism(s) specify the location of AJ, in the absence of apical polarity cues? (2) what signal triggers the onset of rosette formation? (3) do cell–matrix interactions and vascular signals contribute to these processes?

β-catenin has dual functions in the zG, acting as an integral structural molecule of the AJ and as an effector of canonical Wnt signaling in the cytoplasm and nucleus[26]. Our genetic β-catenin loss- and gain-of-function models cannot distinguish between these distinct functions. In the absence of β-catenin we observed loss of F-actin signal, consistent with the role of β-catenin as a linker protein between α-catenin and the actin cytoskeleton. We find that other AJ components, such as K-cadherin, were also absent, suggesting a complete disruption of AJs in these cells. This is consistent with the fact that AJ stability is heavily dependent on the mechano-tension provided by actomyosin[52]. In addition, rosette frequency markedly decreases in these mice, consistent with our hypothesis that rosette formation requires AJ-mediated membrane constriction. However, we cannot rule out the possibility that some aspect of the phenotype is due to disrupted Wnt signaling. Future studies will therefore focus on deletion of AJ-specific proteins such as α-catenin, K-cadherin, and/or N-cadherin. On the other hand, in the β-catenin GOF model, zG expansion is accompanied by an increase in rosette frequency, suggesting β-catenin is a strong driver of zG morphogenesis. However, the extent to which this is due to β-catenin's role in AJs or Wnt signaling remains to be determined.

Canonical Wnt signaling is widely involved in the development and morphogenesis of embryonic tissues[53,54]. It activates a broad transcriptional program that regulates numerous aspects of organogenesis, ranging from proliferation to cell fate specification[55]. It can also regulate morphogenesis by inducing paracrine factors that directly affect cell migration, adhesion, and polarity, such as the FGF signaling pathway[56,57]. There are many examples in development where Wnt/β-catenin signaling induces the expression of FGF ligands and/or receptors, and in turn FGF signaling can inhibit or potentiate the level of canonical Wnt signaling[58,59]. In the current study, we have identified numerous transcripts enriched in βCat-GOF adrenals by RNAseq. These transcripts may or may not be direct transcriptional targets of β-catenin, but nonetheless they provide interesting hypotheses regarding the role of β-catenin in zG morphogenesis. One of these genes is Fgfr2. Our single molecule in situ hybridization data show that the expression domain of Fgfr2 overlaps with the area of Wnt activation, namely the zG in control and the expanded zG in βCat-GOF adrenals. This is consistent with previous reports showing that the Fgfr2 expression pattern in the fetal adrenal is localized to the outer cortex, where Wnt/β-catenin signaling is active[29,60,61]. During neuromast assembly in the lateral line primordium of zebrafish, FGF signaling plays an essential role in

rosette formation by triggering Shroom3-mediated apical constriction[16]. Our data suggest that in the adrenal cortex Shroom3 expression is modulated by FGFR2. Upon Fgfr2 deletion, zG rosette frequency markedly decreases due to defective AJ aggregation. In addition, expression of N- and K-cadherin is also markedly reduced, suggesting FGFR2 may have a general role in regulating cell adhesion in the zG. We find Fgfr2 deletion has no effect on canonical Wnt activity, placing the FGF pathway downstream of Wnt/β-catenin in this context. Formal proof of this hypothesis will require deletion of Fgfr2 in βCat-GOF mice. In addition to FGF signaling, the noncanonical Wnt/PCP pathway also plays a prominent role in tissue morphogenesis during development[62]. It will be interesting to investigate what role noncanonical Wnt signaling might play in zG rosette dynamics.

The significance of rosette-based adrenal glomerular morphology remains to be further elucidated. Here we show that in two distinct mouse models where rosette numbers are either amplified or reduced, the robustness of aldosterone output is affected in the same direction. In β-catenin GOF mice where the morphological zG is greatly expanded, plasma aldosterone levels are significantly higher than control mice. In contrast, Fgfr2-LOF mice, whose adrenals have lost most rosette structures, maintain similar aldosterone output as controls at the expense of higher plasma renin activity. These findings are consistent with a central role for rosettes in the regulation of zG cell function. However, further validation of this model and pinpointing its underlying mechanism(s) will require extensive studies.

One possible function of rosette-based glomerular morphogenesis is that segmentation of groups of cells into distinct glomeruli effectively increases the surface area of the BM, and hence increases exposure to the vasculature. As a key component of the renin–angiotensin–aldosterone system, the adrenal zG is a crucial regulator of $Na^+/K^+$ balance and blood pressure. Hence, there is a constant need for zG cells to monitor and rapidly respond to the circulating levels of Angiotensin II (Ang II) and interstitial $K^+$. Having an extensive epithelial–vascular interface will likely enhance the sensitivity of zG cells to these stimuli and the efficiency of aldosterone release. Interestingly, a recent study by Guagliardo and colleagues revealed zG cells residing within the same rosettes exhibit a highly coordinated $Ca^{2+}$ firing activity pattern, showing that the rosette may serve as an organizing center where zG cell response to stimulant can be synchronized and amplified[63]. However, to determine what causes such cooperativity and whether it directly contributes to robust zG function will require extensive future studies.

Our data show that rosettes are present in the adult zG across species, suggesting a potentially conserved role in on-going adult tissue maintenance and function. Since the adrenal cortex self-renews throughout life, it is likely that rosettes resolve as cells transit from zG to zF and new rosettes form to replace those lost. While this remains to be formally demonstrated, our work provides a conceptual framework to tackle such intriguing questions.

## Methods

**Ethical compliance.** We have complied with all relevant ethical regulations relevant for animal testing and research.

**Mice.** All animal procedures were approved by Boston Children's Hospital's Institutional Animal Care and Use Committee. Mouse strains used in this study were: $AS^{Cre}$ (Cyp11b2$^{tm1.1(cre)Brlt}$)[24], Ctnnb1$^{fl}$ (Ctnnb1$^{tm2Kem}$)[64], Ctnnb1$^{fl(ex3)}$ (Ctnnb1$^{tm1Mmt}$)[65], and Fgfr2$^{fl}$ (Fgfr2$^{tm1Dor}$)[66]. To generate bigenic $AS^{Cre/+}$:: Ctnnb1$^{fl/fl}$ mice (referred to as βCat-LOF), $AS^{Cre/+}$ mice were bred with Ctnnb1$^{fl/fl}$ mice. To generate bigenic $AS^{Cre/+}$:: Ctnnb1$^{fl(ex3)/+}$ mice (referred to as βCat-GOF), $AS^{Cre/+}$ mice were bred with Ctnnb1$^{fl(ex3)/+}$ mice. To generate bigenic $AS^{Cre/+}$:: Fgfr2$^{fl/fl}$ mice (referred to as Fgfr2-LOF), $AS^{Cre/+}$ mice were bred with Fgfr2$^{fl/fl}$ mice. All animals were maintained on a mixed sv129-C57Bl/6 genetic background, with ad lib access to food and water, under a 12-h light/12-h dark cycle. Mice were fed HS (8% NaCl), NS (0.575% NaCl), or LS (0.175% NaCl) chow (ScottPharma

Solutions). Polymerase chain reaction (PCR) was used for genotyping. Littermates were used whenever possible and both male and female animals were studied. All experiments on adult wild type, βCat-LOF, βCat-GOF, Fgfr2-LOF, and respective control animals were carried out at 6–11 weeks of age. Analyses of postnatal morphogenesis were conducted at postnatal days 0, 12, and 3 weeks of age.

**Tissue preparation.** After dissection, adrenals were trimmed of surrounding fat tissue and rinsed in phosphate-buffered saline (PBS). For immunofluorescence, adrenals were cut into halves with a surgical blade and fixed in 4% paraformaldehyde (PFA) at 4 °C overnight. For RNAscope in situ hybridization, intact adrenals were fixed in 4% PFA at room temperature (RT) overnight.

**Floating section immunofluorescence.** After fixation, adrenals were embedded in 4% low-melting-temperature SeaPlaque Agarose (Lonza, 50100) and sectioned at thickness of 100 μm with a vibratome. Individual slices were separated by free floating in PBS. Slices were washed with 0.1% Tween-20 in PBS for 10 min for three times, and blocked in 5% normal goat serum, 1% bovine serum albumin, 0.1% Tween-20 in PBS for 1 h at RT with gentle rocking. Slices were incubated with primary antibodies (1:100) diluted in blocking solution at 4 °C overnight. After three 20 min washes with 0.1% Tween-20 in PBS, slices were incubated with secondary antibodies (1:100) diluted in 0.1% Tween-20 in PBS for 2 h at RT. For F-actin staining, Alexa Fluor 647-conjugated Phalloidin (Invitrogen, A22287) was added to the secondary antibody mix for the last 30 min at final concentration of 1:100. For nuclear staining, DAPI (4′,6-diamidino-2-phenylindole) was added to secondary antibody mix for the last 5 min at final concentration of 1:500–1:1000. Slices were then washed with 0.1% Tween-20 in PBS for 20 min for three times, processed for tissue clearing or directly mounted on Superfrost Plus slides (Fisher Scientific, 12-550-15) with ProLong Gold Antifade Mountant (Thermo Fisher Scientific, P36930). Primary antibodies used for this application include: Rat anti-Laminin β1 (Santa Cruz, sc-33709), Rabbit anti-Gαq (Abcam, ab75825), Rabbit anti-β-catenin (Abcam, ab16051), Rabbit anti-Col4a1 (Novus Biologicals, NB120-6586), Rat anti-CD31 (BD Bioscience, 557355), Rabbit anti-Vimentin (Abcam, ab92547), Rabbit anti-N-cadherin (Novus Biologicals, NBP2-38856), Rabbit anti-K-cadherin (Abcam, ab133632), and Rat anti-E-cadherin (Abcam, ab11512). The following secondary antibodies were used: Alexa Fluor 647/594/488-conjugated goat anti-rabbit IgG, Alexa Fluor 647/594-conjugated goat anti-rat IgG (Invitrogen). Images were acquired using an upright Zeiss LSM710 or LSM700 confocal microscope with either a 10×/0.3 EC Plan-Neofluar, a 40×/1.4 oil Plan-Apochromat or a 63×/1.4 oil Plan-Apochromat objective and adjusted for brightness and contrast in ImageJ.

**Tissue clearing.** After immunostaining, floating 100 μm-thick slices were dehydrated by an ethanol gradient and washed in 100% ethanol twice for 5 min. In a 10-well glass staining dish (Electron Microscopy Sciences, 71564), slices were washed with 50% ethanol in BABB solution (benzyl alcohol: benzyl benzoate (Sigma-Aldrich) in 1:2 ratio (v/v)) for 5 min at RT. Slices were then incubated in fresh BABB solution for 10 min at RT and mounted on Superfrost Plus slides (Fisher Scientific, 12-550-15) with BABB. Cover glass (0.16–0.19 mm in thickness) were applied and sealed with silicone grease (Dow Corning, 1597418).

**Paraffin section immunofluorescence.** After fixation, adrenals were dehydrated in ethanol, xylene, and embedded in paraffin blocks. Paraffin sections were cut at 5 μm thickness. Sections were rinsed in xylene, an ethanol gradient and then PBS. Antigen retrieval was performed in Tris-EDTA pH 9.0 (for Lef1) or 10 mM Sodium Citrate pH 6.0 (all others). Sections were blocked in 5% Normal Goat Serum, 0.1% Tween-20 in PBS for 1 h at RT. Primary antibodies were diluted 1:200 in 5% NGS in PBS and incubated on sections at 4 °C overnight. Slides were washed three times for 5 min in 0.1% Tween-20 in PBS. Secondary antibodies were diluted in 1:200 in PBS and incubated on sections at RT for 1–2 h. For nuclear staining, DAPI (4′,6-diamidino-2-phenylindole) was added to secondary antibody mixture at a final concentration of 1:1000. After three 5-min washes with 0.1% Tween-20 in PBS, slides were mounted with ProLong Gold Antifade Mountant (Thermo Fisher Scientific, P36930). Primary antibodies used for this application include: Mouse anti-β-catenin (BD Biosciences, 610153), Rabbit anti-K-cadherin (Abcam, ab133632), Rabbit anti-Lef1 (Abcam ab137872), and Rabbit anti-Cyp11b2 (kindly provided by Dr. Celso E. Gomez-Sanchez). The following secondary antibodies were used: Alexa Fluor 647-conjugated goat anti-rabbit IgG, Alexa Fluor 594-conjugated goat anti-mouse IgG (Invitrogen). Images were acquired using a Nikon upright Eclipse 90i microscope with a 20×/0.75 Plan-Apochromat objective and adjusted for brightness and contrast in ImageJ.

**Human and peruvian mouse adrenal samples.** The human adrenal sample was collected from a male patient identified as having an adrenal incidentaloma on abdominal imaging. The subject gave written informed consent for histological investigation. The diagnosis of an adrenocortical adenoma was histologically confirmed after surgical resection. Analysis of tissue in this report was limited to neighboring normal adrenal tissue present in the tissue block. The Peruvian mouse adrenal sample was a gift from Drs. Celso Gomez-Sanchez and Elise Gomez-Sanchez. In both cases, paraffin blocks were cut into 10–14 μm-thick sections to obtain maximum spatial information of zG morphology and processed as described above.

**Image analysis.** Quantifications of glomerular roundness, glomerular 2D area and Lamb1-positive zG thickness were performed on confocal images of 100 μm-thick sections stained with Lamb1 in ImageJ. Outlines of individual glomeruli were hand-traced following Lamb1 signal. Gaps in Lamb1 signal at glomerular connections were closed if smaller than 5 μm. Gaps in Lamb1 signal at zG/zF boundary were closed based on the termination of Gαq signal and morphological boundary of zG. Glomerular roundness measurements were obtained using the built-in shape descriptor function of ImageJ, where roundness is calculated as $4 \times area/\pi \times (major\ axis)^2$. Glomerular 2D area measurements were calculated as $\mu m^2$. Lamb1-positive zG thickness measurements were obtained by drawing a transverse line across the Lamb1-positive region. Quantifications of rosette frequency, F-actin cluster size and length were performed on confocal Z-stack images of Phalloidin-stained 100 μm-thick sections. For each Z-stack, analysis was confined to a pre-defined $160 \times 160 \times 10\ \mu m^3$ cortical area oriented so that the horizontal direction is parallel to the capsule. Rosette centers were identified by satisfaction of two criteria: a greater-than-5-cell contact point and the presence of F-actin aggregate at the contact point. Rosette frequency was reported as the number of rosette centers per $160 \times 160 \times 10\ \mu m^3$ stack. Measurements of F-actin cluster size and length were obtained from volume renderings of Z-stacks of Phalloidin-stained sections in Imaris (Bitplane, v7.6.4). Automatic thresholding with background subtraction (local contrast) was applied to each Z-stack for 3D rendering. F-actin cluster size was expressed in $\mu m^3$. F-actin cluster length was calculated as the maximal axial length of the object-oriented minimum Bounding Box statistical variable. For 3D reconstruction of a rosette, outlines of individual cells were hand-traced based on β-catenin staining signal and volume-rendered in Imaris (Bitplane, v7.6.4).

**RNA isolation and RNA sequencing.** Total RNA was isolated from whole adrenals trimmed of adherent fat and homogenized in TRI Reagent (Sigma) using a Direct-zol RNA miniprep kit (Zymo Research), following the manufacturer's protocol. RNA quality was evaluated using an Agilent 2100 Bioanalyzer (Agilent Technologies, Santa Clara, CA), and only samples with RIN > 7.0 were used for RNA sequencing (RNA-seq). Library preparation and sequencing were performed as previously described[67]. Data from all samples were processed using an RNA-seq pipeline implemented in the bcbio-nextgen project (bcbio-nextgen,0.9.1a-b73c090). The following steps were performed by the bcbio pipeline. Adapter sequences were trimmed from reads using cutadapt (version 1.13), the trimmed reads were aligned to the mm10 genome using STAR (version 2.4.1d)[68], and gene expression (uniquely mapping reads) was quantified using featureCounts (version 1.4.4)[69]. To determine which genes were differentially expressed between the control and mutant mice, the count matrix generated by featureCounts was used as input to DESeq2 (version 1.6.3)[70]. To identify differentially expressed genes, the following cutoffs were applied: fold change > 1.4, q value < 0.05, base mean expression > 100. Upregulated gene list was submitted to DAVID (v6.7) for functional annotation enrichment analysis.

**Gene expression analysis.** Total RNA was isolated from whole adrenals as described above, using a Direct-zol microprep kit (Zymo Research). Five hundred nanogram of RNA was reverse-transcribed into cDNA using the High-Capacity cDNA Reverse Transcription Kit (Applied Biosystems, 4368814). Gene expression analysis was performed by quantitative Real Time PCR (qRT-PCR) using a QuantStudio 6 Flex thermocycler (Applied Biosystems). The following Taqman gene expression assays (Applied Biosystems) were used: Axin2 (Mm00443610_m1), Cdh2 (Mm01162497_m1), Cdh6 (Mm01310024_m1), Lef1 (Mm00550265_m1), Shroom3 (Mm00497207_m1), Ppib (Mm00478295_m1), Sulf1 (Mm00552283_m1), Nkd1 (Mm00471902_m1), Nfatc4 (Mm00452375_m1), Shh (Mm00436528_m1), Sema5a (Mm00436500_m1), Daam2 (Mm01273811_m1), Ajuba (Mm00495049_m1), Dact1 (Mm00458117_m1), Wnt4 (Mm01194003_m1), Dab2 (Mm01307290_m1), Gli3 (Mm00492337_m1), Tcf7 (Mm00493445_m1), Fzd5 (Mm00445623_s1), Lama5 (Mm01222029_m1), Prickle1 (Mm01297035_m1). Ppib transcripts were used as the endogenous control and data were expressed using the $2^{-ddCt}$ method.

**Single-molecule RNA in situ hybridization.** After fixation, adrenals were embedded in paraffin blocks and sectioned at 5 μm thickness. Single-molecule RNA in situ hybridization was performed using a RNAscope 2.5 HD Brown Reagent Kit (Advanced Cell Diagnostics, 322300), following manufacturer's protocol. Target retrieval was performed for 7 min[27]. Slides were counter-stained with 30% Gill's Hematoxylin (Fisher Scientific, 23-245654) and mounted with Cytoseal XYL (Fisher Scientific, 22-050-262). The following probes were used: Fgfr2 (ACD, 454951), Shroom3 (ACD, 472221). Positive control Ppib (ACD, 313911) and negative control dapB (ACD, 310043) probes were used in every experiment to ensure sample quality.

**Transmission electron microscopy.** Adrenals were fixed in 2.5% glutaraldehyde, 1.25% paraformaldehyde, and 0.03% picric acid in 0.1 M sodium cacodylate buffer

(pH 7.4) for 2 h at RT, washed in 0.1 M cacodylate buffer and post-fixed with 1% Osmiumtetroxide (OsO4)/1.5% Potassiumferrocyanide (KFeCN6) for 1 h. Samples were washed in water twice, in 50 mM Maleate buffer pH 5.15 (MB) once, and incubated in 1% uranyl acetate in MB for 1 h followed by 1 wash in MB, 2 washes in water and subsequent dehydration in an ethanol gradient (10 min each; 50%, 70%, 90%, 2 × 10 min 100%). Samples were then put in propyleneoxide for 1 h and infiltrated overnight in a 1:1 mixture of propyleneoxide and TAAB Epon (TAAB Laboratories Equipment, Aldermaston, England). On the following day, samples were embedded in TAAB Epon and polymerized at 60 °C for 48 h. Ultrathin sections (about 80 nm) were cut on a Reichert Ultracut-S microtome, picked up on to copper grids stained with lead citrate and examined in a JEOL 1200EX Transmission electron microscope or a TecnaiG² Spirit BioTWIN and images were recorded with an AMT 2k CCD camera.

**Western blotting.** Adrenal lysates were prepared by homogenizing in lysis buffer (50 mM Tris pH 7.5, 150 mM NaCl, 1 mM EDTA, 1% NP40, 0.5% sodium deoxycholate, 1.0% sodium dodecyl sulfate (SDS), 2 mM NaF, 2 mM sodium orthovanadate, and supplemented with protease and phosphatase inhibitors), sonicating and centrifuging at 13,000g for 10 min. Lysates were subjected to SDS polyacrylamide gel electrophoresis and transferred onto a polyvinylidene fluoride membrane (Thermo Scientific). After blocking, blots were incubated overnight with primary antibodies. Secondary antibody conjugated with horseradish peroxidase and chemiluminescent ECL substrate (Bio-Rad, 1705060) were used to develop blots. Antibodies used for this application are: Rabbit anti-FGFR2 clone D4L2V (Cell Signaling Technology, 23328, 1:2000), Rabbit anti-β-Actin clone 13E5 (Cell Signaling Technology, 4970, 1:5000), HRP-linked goat anti-rabbit IgG (Cell Signaling Technology, 7074, 1:5000). Unprocessed scans of blots are provided in a Source Data file.

**Hormone analysis.** Plasma was collected by rapid retro-orbital blood collection with hematocrit tubes. Plasma aldosterone levels were determined using an aldosterone $I^{125}$ radioimmunoassay (Tecan, Morrisville, NC, MG13051). Plasma renin activity was determined on EDTA-collected plasma using a competitive radioimmunoassay for Angiotensin I following the manufacturer's instructions (Tecan, Morrisville, NC, 30120823). To rule out the influence of limiting amounts of angiotensinogen substrate in the plasma, an excess (about 180 ng) of angiotensinogen 1–14 renin substrate porcine (Sigma, SCP0021) was added to every sample, calibrator and control right before quantification. Gamma counts were measured for 1 min.

**Quantification and statistical analysis.** All statistical analyses and graphs were generated in R (v3.3.2) or Prism 7 (GraphPad Software). Nonparametric tests were applied to glomerular morphometric data, including Mann–Whitney U test and Kruskal–Wallis one-way ANOVA on ranks with Dunn's multiple comparison correction. Unpaired two-tailed t tests were used to analyze rosette frequency, zG thickness, and qRT-PCR data, using the Holm–Sidak method for multiple comparison correction. $P < 0.05$ was considered statistically significant. For all box and whisker plots, box boundaries are the 25th and 75th percentiles, and the whiskers are the 5th and 95th percentiles. For all bar graphs, error bars represent SEM. Exact values of $n$ are reported in each figure legend.

**Reporting summary.** Further information on research design is available in the Nature Research Reporting Summary linked to this article.

## Data availability

Sequencing data have been deposited in GEO with the accession code GSE144503. Source data underlying Fig. 3d–g, 4b–g, 5b–g, 6b–d, 7b–g, and Supplementary Figs 2b, 5b–d, 7b–e, 8c, and 9a, c are provided as a Source Data file. All other data supporting the findings of this study are available from the corresponding author upon reasonable request.

## Code availability

All R scripts used in data analysis and generation of figures are available upon request.

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

## Acknowledgements

We thank members of the Breault Lab and Dr. Joseph A. Majzoub for helpful discussion and comments on the paper. We also thank Dr. Hisashi Umemori for the gift of *Fgfr2^{fl/fl}* mice and Dr. Celso Gomez-Sanchez and Dr. Elise Gomez-Sanchez for Peruvian mouse adrenal samples. We thank the Dana–Farber Molecular Biology Core, the Harvard Chan Bioinformatics Core, the IDDRC Cellular Imaging Core, the HMS Electron Microscopy Facility and the BIDMC Histology Core for providing excellent technical expertize. Work by R.S.K. at the Harvard Chan Bioinformatics Core and Center for Stem Cell Bioinformatics was supported by funding from the Harvard Stem Cell Institute. This work was also supported by IDDRC P30HD18655, R01-DK100653, and R01-DK123694 to D.T.B. and by the German Research Foundation (DFG) project number 314061271 (CRC/TRR 205) to F.B.

## Author contributions

S.L., P.Q.B., and D.T.B. conceptualized the study; S.L., E.P., and D.T.B. developed the methodology; S.L., E.P., M.S.S., R.S.K., and S.X. performed the experiments; F.B. and M.M.T. provided the essential reagents; all authors interpreted the data; S.L., D.L.C., and D.T.B. wrote the original paper with editorial inputs from all authors; J.M., D.L.C., and D.T.B. provided the supervision; D.T.B. acquired the funds.

## Competing interests

The authors declare no competing interests.
