## [Peer Review File · Nature Communications]

Reviewers' comments:

Reviewer #1 (Remarks to the Author):

The manuscript by Leng S., et al describes the role of beta catenin and FGFR2 in the morphogenesis of the adrenal cortex Zona Glomerulosa (ZG), specifically in the formation of "rosette"-like structures using mouse models. A thorough morphometric analysis is followed by examination of ZG from beta-catenin loss of function (LOF) and gain of function (GOF) mice, a transcriptome comparative analysis between wild-type and GOF (leading to the discovery of FGFR2), followed by functional investigation of FGFR2 using a ZG specific FGFR2 LOF mouse. The manuscript is beautifully written and the images are of very high quality and convincing, as well as the analysis (of data and images) and the associate stats have been carried out properly in my opinion. The videos are particularly helpful to appreciate ZG structure with regards to ECM markers. Overall, the data presented add novelty to the field, will undoubtedly further our understanding on the complex pathways' interactions in the adrenal cortex subcapsular region and represents the first high- quality and detailed morphometric analysis of the ZG.

In my opinion the manuscript would benefit from some extra experiments aimed at further characterising rosettes in relation to the known heterogeneity of the ZG. Furthermore, the readership of such manuscript would be much wider if there was a more complete characterization of rosette-like structure in human adrenal cortices.

1) The mouse ZG is usually composed by SF1-positive CYP11B2-positive cells, however clusters of SF1-positive CYP11B2 negative/SHH positive (precursors) cells are interspersed within the ZG. It is acknowledged that beta catenin cannot discriminate these two ZG populations, while SHH and CYP11B2 can in most cases. Would the authors be in a position to assess whether any difference in rosette structure exists between precursor and functional ZG cells in mice? Moreover, if SHH precursor cells also form rosette-like structures then LOF beta-catenin mice (which, having an AS promoter, should not affect SHH clusters) should have a preponderance of intact "SHH-rosettes" (unless somehow affected by adjacent "AS-rosettes"). This experimental set-up can be done using a SHH-lacZ mouse (it is rather recognised that there are not great antibodies to SHH, or any of the SHH pathway components), or double staining with their markers together with CYP11B2 staining or RNAScope, for which they have the expertise (see Fig 6E).

2) The authors show staining for beta-catenin in one human adrenal sample inferring that rosette-like structures are also present in the human adrenal cortex. The human adrenal cortex remodels to generate clusters of cells highly expressing CYP11B2 (AS, named aldosterone producing cell clusters, APCCs) and these clusters might represent one precursor of aldosterone-producing adenomas (APAs). Other clusters structures containing progenitor-like cells expressing the Notch atypical ligand *dlk1* have also been described very recently. A typical adult (post-pubertal) physiologically normal human adrenal section would therefore contain large areas of histological ZG which is largely CYP11B2-negative, some classic (layered-continuous) CYP11B2 staining and APCCs scattered around, the same for *dlk1* albeit in an exclusive manner compare to B23 staining. Are rosette-like structures any different between these ZG areas (proper ZG, ZG which is CYP11B2 negative and APCCs)? Does this change during the lifetime (i.e. how old is the donor of the section used in Fig S 2C ? Is the beta-catenin staining any different across the ZG?. Are they present in late gestational human foetal adrenals? Overall, it would be useful to the readers to have a better morphometric analysis of human samples. This reviewer acknowledges that obtaining such samples can be tedious.

3) While Lamb1 staining is very much ZG-specific, some information (i.e. high mag images) on the pattern of beta-catenin (low levels) F-actin, cadherins, vimentin in the ZF while discussing differences would be useful to the reader. This can be in the Supplemental. How are AJ in the ZF?

4) Aldosterone (corticosterone?) levels in LOS and GOF mice, can it be measured?

5) The ZG can remodel rather quickly upon dietary or pharmacological intervention. The ZG will disappear on a salty diet (or ACE inhibitor) and will considerably expand on a diet very low in sodium. It would be interesting to assess the changes in rosette (number, appearance, roundness

index) during ZG expansion.

Reviewer #2 (Remarks to the Author):

The adrenal zona glomerulosa (zG), the source of aldosterone, undergoes constant turnover. Stem cells in the adrenal capsule differentiate and migrate centripetally to populate the zG, while fully differentiated cells exit zG by direct lineage conversion into glucocorticoid-producing zona fasciculata cells. The zG can expand or contract in response to physiological or pharmacological stimuli.

In this manuscript, Leng et al perform a detailed morphological, biochemical, and genetic analysis of glomeruli within the mouse zG, so as to gain insight into the formation and homeostasis of this zone. By subjecting 100 μm sections of adult adrenal to confocal microscopy, the authors show that glomeruli are enveloped by a Lamb1-rich basement membrane. Within a typical glomerulus, 10-15 zG cells are connected at a common membrane contact point in the center of the structure, thus forming a multicellular rosette. β -catenin, N-cadherin, K-cadherin, and aggregates of F-actin co-localize at the rosette centers and at small dispersed punctae, presumed to represent adherens junctions (AJs) and associated actin- and myosin-containing filaments. Transmission EM confirms the presence of AJs at rosette centers, suggesting a role for AJ-mediated membrane constriction in the formation of these structures. By analyzing P0, P12, and P21 mice, the investigators demonstrate that glomeruli mature postnatally and that this coincides with rosette formation, implying that rosette formation may be integral to glomerular morphogenesis and tissue remodeling. Conditional deletion of *Ctnnb1* (encoding β -catenin) within zG cells disrupts AJ formation and decreases rosette number. On the other hand, constitutive stabilization of β -catenin in zG cells results in increased AJs and more rosettes; RNA-seq and confirmatory qPCR and ISH of these adrenals shows enrichment for regulators of epithelial morphogenesis, including *Fgfr2* and *Shroom3*. Conditional deletion of *Fgfr2* results in decreased AJs, reduced rosettes, and decreased expression of *Shroom3*. Interestingly, canonical Wnt signaling activity (e.g., *Lef1*, *Axin2*) is unaffected. The authors conclude that: 1) β -catenin and FGFR2 regulate rosette-based adrenal morphogenesis, and 2) rosettes control adult tissue remodeling.

This is an outstanding manuscript that redefines our perception of adrenocortical anatomy, development, and homeostasis. The data are compelling. The figures and videos are beautiful. The statistical analysis appears sound. The work will appeal to a broad audience including developmental biologists, cell biologists, and clinical endocrinologists.

In a companion manuscript, Guagliardo et al ("Coordinated Calcium Burst Firing within the Aldosterone-Producing Adrenal Rosettes") show that cells within rosettes functionally interact to produce synchronized bursts of activity in response to angiotensin II. This extends the findings of the current manuscript and reinforces its fundamental importance.

Minor comments:

1. Lines 207 and 227. The authors note phenotypic differences between female and male mice. They may wish to cite recent articles on the topic of sexual dimorphism in the mouse adrenal (e.g., PMID: 31104943, PMID: 29367455)
2. Line 305. For consistency, abbreviate "adherens junctions" as "AJs."

Reviewer #3 (Remarks to the Author):

The manuscript by Leng et al. investigates morphogenesis of zona glomerulosa (zG) during development. They discovered that cells in this region of the adrenal gland are organized into rosettes in both mouse and human. Using a mouse model, they show that these rosettes form postnatally through the constriction of adherence junction. They provide a careful analysis of rosettes including 3D reconstruction and EM approaches. To address the molecular bases of rosette formation, they manipulate b-catenin levels to show that b-cat is necessary for proper rosette size and structure. To investigate genes that function downstream of b-cat they perform RNA-seq experiment comparing wild-type adrenals to b-cat GOF. They found that one of the genes regulated by b-cat was FGFR2. The authors generated a tissue-specific FGFR2 KO and demonstrated that FGFR2 is required for adrenal rosette formation by regulating AJ abundance and aggregation. They also suggest that FGFR2 regulates Shroom3, a scaffold protein, that in other system is necessary for rosette formation. I appreciate a novelty of this study demonstrating rosettes as a functional postnatal structure; this is different from many other systems during development where rosettes are formed transiently during tissue morphogenesis. The authors perform careful structural and time-course analysis of rosette formation. Overall, this work will be of general interest to scientists studying tissue and organ morphogenesis. However, the paper can be significantly strengthened by connecting rosette morphogenesis to adrenal function.

Specific comments:

The authors claim that this is the first example of adult rosettes. This is not quite the case, as rosettes are present in the ventricular zone of the adult mammalian brain.

One of the main deficiencies of the paper is that it is not clear what is the functional significance of rosettes in the adult adrenal. The authors speculate that these structures may increase exposure to the vasculature. However, they do not assay how their manipulation (B-cat LOF and GOF as well as FGFR2 KO) affects adrenal function in these mouse models.

The authors found smaller or larger rosettes in b-cat LOF and b-cat GOF, respectively. Do these have comparable numbers of cells? Similarly, the time course of these loss- and gain-of-function models would provide information when these pathways are important and their roles in cellular mechanism of rosette formation.

The actin staining in Fig 3C is hard to see; it should be shown as a separate channel.

Reviewer #1 (Remarks to the Author):

The manuscript by Leng S., et al describes the role of beta catenin and FGFR2 in the morphogenesis of the adrenal cortex Zona Glomerulosa (ZG), specifically in the formation of “rosette”-like structures using mouse models. A thorough morphometric analysis is followed by examination of ZG from beta-catenin loss of function (LOF) and gain of function (GOF) mice, a transcriptome comparative analysis between wild-type and GOF (leading to the discovery of FGFR2), followed by functional investigation of FGFR2 using a ZG specific FGFR2 LOF mouse. The manuscript is beautifully written and the images are of very high quality and convincing, as well as the analysis (of data and images) and the associate stats have been carried out properly in my opinion. The videos are particularly helpful to appreciate ZG structure with regards to ECM markers. Overall, the data presented add novelty to the field, will undoubtedly further our understanding on the complex pathways' interactions in the adrenal cortex subcapsular region and represents the first high-quality and detailed morphometric analysis of the ZG.

In my opinion the manuscript would benefit from some extra experiments aimed at further characterising rosettes in relation to the known heterogeneity of the ZG. Furthermore, the readership of such manuscript would be much wider if there was a more complete characterization of rosette-like structure in human adrenal cortices.

- 1. The mouse ZG is usually composed by SF1-positive CYP11B2-positive cells, however clusters of SF1-positive CYP11B2 negative/SHH positive (precursors) cells are interspersed within the ZG. It is acknowledged that beta catenin cannot discriminate these two ZG populations, while SHH and CYP11B2 can in most cases. Would the authors be in a position to assess whether any difference in rosette structure exists between precursor and functional ZG cells in mice? Moreover, if SHH precursor cells also form rosette-like structures then LOF beta-catenin mice (which, having an AS promoter, should not affect SHH clusters) should have a preponderance of intact “SHH-rosettes” (unless somehow affected by adjacent “AS-rosettes”). This experimental set-up can be done using a SHH-lacZ mouse (it is rather recognised that there are not great antibodies to SHH, or any of the SHH pathway components), or double staining with their markers together with CYP11B2 staining or RNAScope, for which they have the expertise (see Fig 6E).*

Response: We appreciate the reviewer raising this important point. It is widely recognized that the zG is composed of differentiated Cyp11b2+ cells and less mature Cyp11b2- Shh+ progenitor cells. We agree that it is important to understand if these two populations reside within the same rosette-based structures. To address this, we re-examined the cellular composition of zG glomeruli in relation to Cyp11b2 expression in mice. Interestingly, under normal homeostatic conditions, we find a wide range in the percentage of Cyp11b2+ cells in each glomerulus, with the majority varying from 20% to 70%. This suggests that Cyp11b2+ cells are not physically separated from their progenitors by the basement membrane, which may allow progenitors to quickly contribute to glomerular expansion or regeneration upon stress. In light of these new data, we think the rosette disruption phenotype we observe in our β Cat-LOF and Fgfr2-LOF models is largely due to the loss of these genes in Cyp11b2+ cells in each glomerulus. In most glomeruli, this results in a significant disruption due to the multicellular nature of rosette structures. These new data are shown in **Supplemental Fig. 2** and discussed in **lines 132-140** of the revised manuscript. We apologize for not including data demonstrating co-staining between Cyp11b2 and other rosette-defining markers such as F-actin or Lamb1. Unfortunately, the only available Cyp11b2 antibody works in paraffin sections, which is not compatible with phalloidin or Lamb1 staining. In addition, while the alternative approach of combining RNAScope with IHC would be useful in addressing this question, we have found it technically challenging as many epitopes for IHC do not survive the harsh treatment conditions of RNAScope. However, in aggregate we believe that the data presented in the remainder of the manuscript provides sufficient evidence to support a strong correlation among the various markers.

2. *The authors show staining for beta-catenin in one human adrenal sample inferring that rosette-like structures are also present in the human adrenal cortex. The human adrenal cortex remodels to generate clusters of cells highly expressing CYP11B2 (AS, named aldosterone producing cell clusters, APCCs) and these clusters might represent one precursor of aldosterone-producing adenomas (APAs). Other clusters structures containing progenitor-like cells expressing the Notch atypical ligand dlk1 have also been described very recently. A typical adult (post-pubertal) physiologically normal human adrenal section would therefore contain large areas of histological ZG which is largely CYP11B2-negative, some classic (layered-continuous) CYP11B2 staining and APCCs scattered around, the same for dlk1 albeit in an exclusive manner compare to B23 staining. Are rosette-like structures any different between these ZG areas (proper ZG, ZG which is CYP11B2 negative and APCCs)? Does this change during the lifetime (i.e. how old is the donor of the section used in Fig S 2C)? Is the beta-catenin staining any different across the ZG? Are they present in late gestational human foetal adrenals? Overall, it would be useful to the readers to have a better morphometric analysis of human samples. This reviewer acknowledges that obtaining such samples can be tedious.*

Response: We agree with the reviewer that a systematic and thorough morphometric analysis of the human adrenal focusing on the various normal and pathological zG areas from fetal to adult ages will be very interesting to a wide readership. We recognize the importance of this question, and we hope that this manuscript will raise awareness of this issue and lead to a concerted effort to build such a tissue collection within the community of adrenal biologists. We submit that the human data in our manuscript serve as important proof-of-principle evidence that the human adrenal also contain rosette structures, which require future study.

3. *While Lamb1 staining is very much ZG-specific, some information (i.e. high mag images) on the pattern of beta-catenin (low levels) F-actin, cadherins, vimentin in the ZF while discussing differences would be useful to the reader. This can be in the Supplemental. How are AJ in the ZF?*

Response: Thank you for this very helpful suggestion. We have now incorporated more information on the zF including a more detailed description in the Results Section and high magnification images, which are shown in **Supplemental Fig. 1a-c** and **Supplemental Fig. 3a, b** and discussed in **lines 148, 156-159** of the revised manuscript. To summarize, we find that AJ markers such as cadherins and F-actin stain much weaker and appear sparser in the zF. In addition, they do not form punctate aggregates as in zG cells. This suggests that AJs play a less prominent role in zF cell-cell adhesion and tissue organization, and further support that the mechanisms regulating rosettes are specific to the zG. Vimentin is a marker of mesenchymal cells and we show that it is excluded from the zG and zF epithelium. Instead, vimentin strongly stains capsular cells and a rare stromal population in the adrenal cortex.

4. *Aldosterone (corticosterone?) levels in LOF and GOF mice, can it be measured?*

Response: This is indeed a very important question and could potentially speak to the functional significance of rosette structures. To address this point, we have assessed plasma hormone levels in our β Cat-GOF and Fgfr2-LOF models. We find that β Cat-GOF mice have significantly higher plasma aldosterone levels than control mice (**Figure 5g, lines 263-265**). On the other hand, Fgfr2-LOF mice demonstrated similar plasma aldosterone levels as control mice, albeit with a corresponding increase in the level of plasma renin activity (**Figure 7g, lines 333-338**). Taken together, these findings are consistent with a state of compensated hypoaldosteronism. These new data support the concept that rosette number positively correlates with zG physiological function. However, we cannot rule out the possibility that Wnt/ β -catenin and FGF signaling pathways might have other effects on zG cellular

function that are independent of rosette biology. We have not measured corticosterone since our data have shown no histological change in the zF in these models.

5. *The ZG can remodel rather quickly upon dietary or pharmacological intervention. The ZG will disappear on a salty diet (or ACE inhibitor) and will considerably expand on a diet very low in sodium. It would be interesting to assess the changes in rosette (number, appearance, roundness index) during ZG expansion.*

Response: We thank the reviewer for raising this intriguing question. To determine if varying dietary sodium levels impacts zG glomerular morphology, we treated adult mice for one week with a high salt (HS, 8% NaCl), low salt (LS, 0.175% NaCl) or normal salt (NS, 0.575% NaCl) diet. Our results are shown in **Supplement Fig. 5** and discussed in **lines 209-223** of the revised manuscript. We found that one week of sodium manipulation does have a moderate effect on the size of the zG as marked by Lamb1, which corresponds to changes in glomerular 2D area. Specifically, we found that under HS diet, mice have a thinner zG and smaller glomeruli, whereas a LS diet led to a thicker zG with bigger glomeruli. Interestingly, both HS and LS diets cause a decrease in glomerular roundness. The glomeruli in HS-treated mice appear flatter and more compact, whereas LS-treated mice have glomeruli that are more elongated. While no significant change in rosette number was observed, we did detect a trend towards an increase in mice on HS versus LS diet. These new data reveal how dietary sodium can modulate zG glomerular morphology.

Reviewer #2 (Remarks to the Author):

The adrenal zona glomerulosa (zG), the source of aldosterone, undergoes constant turnover. Stem cells in the adrenal capsule differentiate and migrate centripetally to populate the zG, while fully differentiated cells exit zG by direct lineage conversion into glucocorticoid-producing zona fasciculata cells. The zG can expand or contract in response to physiological or pharmacological stimuli.

In this manuscript, Leng et al perform a detailed morphological, biochemical, and genetic analysis of glomeruli within the mouse zG, so as to gain insight into the formation and homeostasis of this zone. By subjecting 100 μm sections of adult adrenal to confocal microscopy, the authors show that glomeruli are enveloped by a Lamb1-rich basement membrane. Within a typical glomerulus, 10-15 zG cells are connected at a common membrane contact point in the center of the structure, thus forming a multicellular rosette. β -catenin, N-cadherin, K-cadherin, and aggregates of F-actin co-localize at the rosette centers and at small dispersed punctae, presumed to represent adherens junctions (AJs) and associated actin- and myosin-containing filaments. Transmission EM confirms the presence of AJs at rosette centers, suggesting a role for AJ-mediated membrane constriction in the formation of these structures. By analyzing P0, P12, and P21 mice, the investigators demonstrate that glomeruli mature postnatally and that this coincides with rosette formation, implying that rosette formation may be integral to glomerular morphogenesis and tissue remodeling. Conditional deletion of Ctnnb1 (encoding β -catenin) within zG cells disrupts AJ formation and decreases rosette number. On the other hand, constitutive stabilization of β -catenin in zG cells results in increased AJs and more rosettes; RNA-seq and confirmatory qPCR and ISH of these adrenals shows enrichment for regulators of epithelial morphogenesis, including Fgfr2 and Shroom3. Conditional deletion of Fgfr2 results in decreased AJs, reduced rosettes, and decreased expression of Shroom3. Interestingly, canonical Wnt signaling activity (e.g., Lef1, Axin2) is unaffected. The authors conclude that: 1) β -catenin and FGFR2 regulate rosette-based adrenal morphogenesis, and 2) rosettes control adult tissue remodeling.

This is an outstanding manuscript that redefines our perception of adrenocortical anatomy, development, and homeostasis. The data are compelling. The figures and videos are beautiful. The statistical analysis

appears sound. The work will appeal to a broad audience including developmental biologists, cell biologists, and clinical endocrinologists.

In a companion manuscript, Guagliardo et al ("Coordinated Calcium Burst Firing within the Aldosterone-Producing Adrenal Rosettes") show that cells within rosettes functionally interact to produce synchronized bursts of activity in response to angiotensin II. This extends the findings of the current manuscript and reinforces its fundamental importance.

Minor comments:

1. Lines 207 and 227. The authors note phenotypic differences between female and male mice. They may wish to cite recent articles on the topic of sexual dimorphism in the mouse adrenal (e.g., PMID: 31104943, PMID: 29367455)

Response: We thank the reviewer for this helpful suggestion. The references have been incorporated in the text on **lines 260-263**.

2. Line 305. For consistency, abbreviate "adherens junctions" as "AJs."

Response: We thank the reviewer for this helpful suggestion. We have edited the manuscript accordingly.

Reviewer #3 (Remarks to the Author):

The manuscript by Leng et al. investigates morphogenesis of zona glomerulosa (zG) during development. They discovered that cells in this region of the adrenal gland are organized into rosettes in both mouse and human. Using a mouse model, they show that these rosettes form postnatally through the constriction of adherence junction. They provide a careful analysis of rosettes including 3D reconstruction and EM approaches. To address the molecular bases of rosette formation, they manipulate b-catenin levels to show that b-cat is necessary for proper rosette size and structure. To investigate genes that function downstream of b-cat they perform RNA-seq experiment comparing wild-type adrenals to b-cat GOF. They found that one of the genes regulated by b-cat was FGFR2. The authors generated a tissue-specific FGFR2 KO and demonstrated that FGFR2 is required for adrenal rosette formation by regulating AJ abundance and aggregation. They also suggest that FGFR2 regulates Shroom3, a scaffold protein, that in other system is necessary for rosette formation. I appreciate a novelty of this study demonstrating rosettes as a functional postnatal structure; this is different from many other systems during development where rosettes are formed transiently during tissue morphogenesis. The authors perform careful structural and time-course analysis of rosette formation. Overall, this work will be of general interest to scientists studying tissue and organ morphogenesis. However, the paper can be significantly strengthened by connecting rosette morphogenesis to adrenal function.

Specific comments:

1. The authors claim that this is the first example of adult rosettes. This is not quite the case, as rosettes are present in the ventricular zone of the adult mammalian brain.

Response: We appreciate the reviewer pointing out this error. We have edited the text accordingly on **line 343**.

2. *One of the main deficiencies of the paper is that it is not clear what is the functional significance of rosettes in the adult adrenal. The authors speculate that these structures may increase exposure to the vasculature. However, they do not assay how their manipulation (B-cat LOF and GOF as well as FGFR2 KO) affects adrenal function in these mouse models.*

Response: To address this important point, we have assessed zG function by measuring aldosterone output in β Cat-GOF and Fgfr2-LOF mice. Please see response to **Reviewer 1 Comment #4**. We have also included more discussion in **lines 415-423**. We acknowledge that it is difficult to directly attribute the observed changes in zG function in these mice to their altered rosette morphology. However, our data demonstrate a clear correlation between rosettes and robust zG function. Furthermore, data presented in a jointly resubmitted manuscript by our collaborators, Nick Guagliardo, Paula Barrett and Mark Beenharkker, highlight the significance of rosettes as a zG cell signaling center using a calcium imaging approach. This manuscript, also under consideration at Nature Communications, is entitled, "**Coordinated Calcium Burst Firing within the Aldosterone-Producing Adrenal Rosette.**" Specifically, their data reveal a high degree of coordination in the calcium firing activity among cells residing within the same rosette, which suggests rosettes play a key role in facilitating cell-cell communication. Hence, together our manuscripts point to rosettes having an instrumental role in regulating zG cell function.

3. *The authors found smaller or larger rosettes in b-cat LOF and b-cat GOF, respectively. Do these have comparable numbers of cells? Similarly, the time course of these loss- and gain-of-function models would provide information when these pathways are important and their roles in cellular mechanism of rosette formation.*

Response: Thank you for this question. To address this, we quantified the number of cells per glomerulus in β Cat-LOF and β Cat-GOF adrenals. These new data are shown in **Figure 4d** and **Figure 5d** and described in **lines 236 and 251** of the revised manuscript. We found that the number of cells correlates with the glomerular cross-sectional area measurements. We agree that a detailed time course study of all the models would provide more insight. However, given that the AS-Cre allele is expressed in a small number of zG cells at birth, which gradually contribute to the entire zG in the first few weeks of life, it is likely that analysis of these mice at early stages will reveal a partial phenotype. Therefore, we respectfully submit that such experiments would be unlikely to yield satisfying results.

4. *The actin staining in Fig 3C is hard to see; it should be shown as a separate channel.*

Response: We thank the reviewer for this helpful suggestion. We have made a separate panel for the F-actin channel alone in **Figure 3c**.

REVIEWERS' COMMENTS:

Reviewer #1 (Remarks to the Author):

The reviewer would like to thank the authors of NCOMMS-19-18490A for carefully taking into consideration my comments and for performing novel experiments in order to assess i) the phenotype of rosettes, ii) AJ structure in the zone fasciculata, iii) aldosterone levels in transgenic mice and iv) rosette structure upon remodelling of the ZG in low/high sodium.

Like the initial submitted manuscript, the new data is of high quality and very informative.

Dr Leonardo Guasti

Reviewer #3 (Remarks to the Author):

In the revised manuscript, the authors carefully considered most of my comments. My concern was addressing how adrenal (or zG more specifically) function is related to zG rosettes. The authors provide evidence, although somewhat correlative, but nevertheless important that rosettes are important for zG function. They show that β Cat-GOF mice have higher plasma aldosterone levels, whereas Fgfr2-LOF animals display higher plasma renin activity. Because rosettes are disrupted in these transgenic animal models it suggests that rosettes are indeed required for correct zG function. Although, the authors acknowledge that both Wnt and Fgf pathways may have other functions in the adrenal, independent of rosette formation. In addition, they pointed out that an accompanied manuscript under review connects rosettes to the calcium signaling in zG. In summary, this point is addressed very well in the revised manuscript.

My other comment was related to the number of cells per rosette in β Cat-GOF and β Cat-LOF manipulations. The authors provide numbers for adults. However, they are unable to look at the number during development for technical reasons.

In summary, I feel that all my comments have been adequately addressed.